# ThermalGen: Style-Disentangled Flow-Based Generative Models for RGB-to-Thermal Image Translation

**Jiuhong Xiao**
New York University
`jx1190@nyu.edu`

**Roshan Nayak**
New York University
`rn2588@nyu.edu`

**Ning Zhang**
Technology Innovation Institute
`ning.zhang@tii.ae`

**Daniel Tortei**
Technology Innovation Institute
`daniel.tortei@hotmail.com`

**Giuseppe Loianno**
University of California, Berkeley
`loiannog@berkeley.edu`

## Abstract

Paired RGB-thermal data is crucial for visual-thermal sensor fusion and cross-modality tasks, including important applications such as multi-modal image alignment and retrieval. However, the scarcity of synchronized and calibrated RGB-thermal image pairs presents a major obstacle to progress in these areas. To overcome this challenge, RGB-to-Thermal (RGB-T) image translation has emerged as a promising solution, enabling the synthesis of thermal images from abundant RGB datasets for training purposes. In this study, we propose **ThermalGen**, an adaptive flow-based generative model for RGB-T image translation, incorporating an RGB image conditioning architecture and a style-disentangled mechanism. To support large-scale training, we curated eight public satellite-aerial, aerial, and ground RGB-T paired datasets, and introduced three new large-scale satellite-aerial RGB-T datasets—**DJI-day**, **Bosonplus-day**, and **Bosonplus-night**—captured across diverse times, sensor types, and geographic regions. Extensive evaluations across multiple RGB-T benchmarks demonstrate that ThermalGen achieves comparable or superior translation performance compared to existing GAN-based and diffusion-based methods. To our knowledge, ThermalGen is the first RGB-T image translation model capable of synthesizing thermal images that reflect significant variations in viewpoints, sensor characteristics, and environmental conditions. Project page: [xjh19971.github.io/ThermalGen](xjh19971.github.io/ThermalGen)

## 1 Introduction

Visual-thermal sensor fusion [1] and cross-modality downstream tasks [29, 50, 57, 58, 42] have become increasingly important for integrating visual and thermal data to achieve robust perception under challenging conditions such as low illumination and adverse weather. While deep learning methods [24] have shown promise in these areas, they typically depend on paired RGB-Thermal (RGB-T) data for training. However, such datasets are limited in availability and often difficult to generalize across applications due to domain gaps. Additionally, the cost of collecting synchronized and calibrated RGB-T data is substantial. These limitations motivate the adoption of generative models to synthesize thermal images from RGB inputs as a more scalable and flexible alternative.

Synthesizing thermal data to construct paired RGB-T datasets brings several advantages for cross-modality downstream tasks. First, this method ensures perfectly aligned details between real RGB images and their synthesized thermal counterparts, providing high-quality data for tasks that de-

39th Conference on Neural Information Processing Systems (NeurIPS 2025).

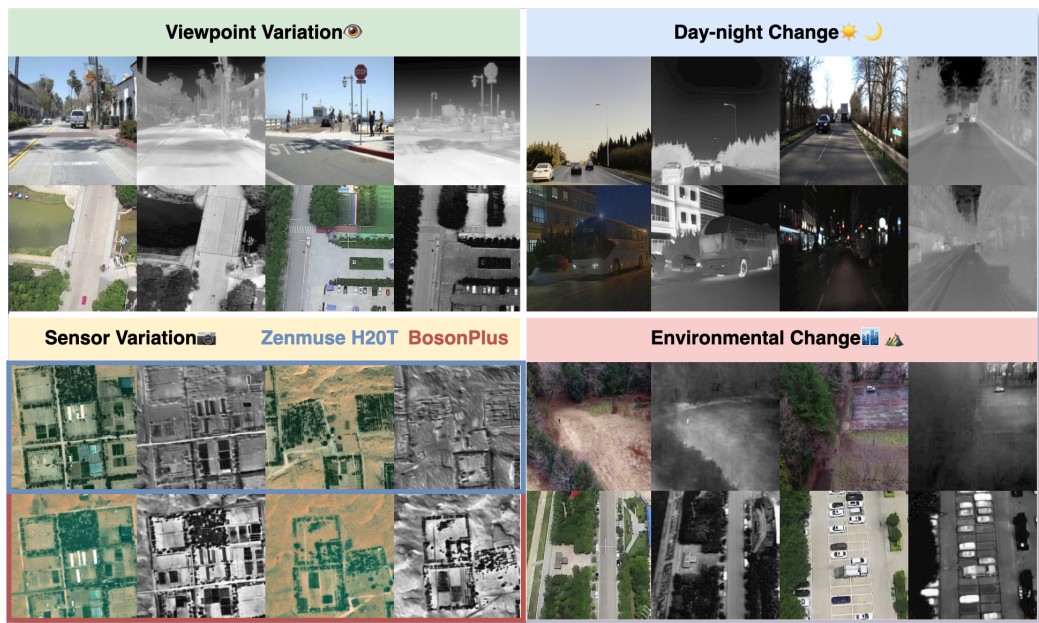

Figure 1: **ThermalGen exhibits robust performance in RGB-to-thermal image translation under diverse conditions.** We present RGB inputs alongside generated thermal images using ThermalGen across a range of challenging variations, including viewpoint variation, day-night change, sensor variation, and environmental change. Variations are illustrated between the two rows in each group.

mand precise cross-modal correspondence, such as dense feature matching [45, 6] and keypoint matching [30]. Second, it enables researchers to exploit the vast repositories of publicly available RGB data, substantially expanding the scale and diversity of training datasets beyond the limited scope of hardware-captured RGB-T pairs, which require specialized calibrated sensors [32, 47] and often suffer from restricted viewpoints [19, 27]. Third, generative models can simulate various thermal characteristics and environmental conditions from a single RGB input, thereby enhancing the robustness of downstream models to variations in thermal sensors and environmental settings.

Benefiting from these advantages, recent studies [57, 58, 56, 50, 20, 12, 10] have demonstrated promising results by leveraging synthesized RGB-T datasets to improve model performance on downstream tasks significantly. In this work, we introduce **an adaptive RGB-to-thermal image translation model** (Fig. 1), aimed at high-fidelity generation across diverse viewpoints, thermal sensors, and environmental factors. The main contributions of this paper are as follows:

- We introduce **ThermalGen**, an adaptive flow-based generative model for RGB-T image translation. Trained on large-scale RGB-T datasets, ThermalGen incorporates an RGB image conditioning architecture with a style-disentangled mechanism, enabling robust generation of thermal images across a wide spectrum of RGB-T styles influenced by thermal sensors, viewpoints, and environmental conditions. The model's architecture facilitates joint training with additional datasets, promoting adaptability across various applications.

- We release three new satellite-aerial RGB-T paired datasets—**DJI-day**, **Bosonplus-day**, and **Bosonplus-night**—comprising aligned satellite RGB and aerial thermal images with variations in time of day, sensor type, and geographic region. Additionally, we curate an extensive collection of public satellite-aerial, aerial, and ground RGB-T datasets to establish a comprehensive benchmark for large-scale training and evaluation.

- Extensive experiments validate ThermalGen's effectiveness across multiple benchmarks, demonstrating comparable or superior performance against both GAN- and diffusion-based baselines. We present comprehensive quantitative metrics and qualitative analyses that illustrate the visual fidelity and style embedding influence of our approach. To our knowledge, ThermalGen represents the first model capable of generating high-fidelity thermal imagery across diverse sensor types, viewpoints, and environmental conditions.

## 2 Related Works

**Synthesized Thermal Data for Downstream Tasks.** Recent studies highlight the growing trend of leveraging synthetic thermal data to enhance RGB-T downstream tasks. XoFTR [50] applies randomized cosine transformations to RGB images for improved cross-modal alignment. MIN-IMA [20] introduces a versatile data engine generating paired datasets across multiple modalities. MatchAnything [12] synthesizes various modalities from inputs like videos and multi-view images for generalizable alignment. AVIID [10] uses synthetic aerial thermal data for object detection, while STGL [57] and STHN [58, 56] boost satellite-thermal retrieval and homography estimation through synthetic RGB-T datasets. These efforts highlight thermal image synthesis as a practical solution to paired data scarcity. In this work, we propose an RGB-T image translation model for general-purpose use, offering broad applicability across diverse downstream tasks.

**RGB-Thermal Image Translation.** Thermal image synthesis from RGB inputs can be formulated as an RGB-T image translation task—an inherently complex problem due to three primary challenges. First, the limited availability of calibrated RGB-T datasets hinders the performance of supervised learning. Unsupervised approaches, such as CycleGAN [63], also struggle to bridge this gap [25]. Second, RGB images inherently lack thermal information, requiring the model to infer thermal cues solely from semantic content. Third, inconsistencies in thermal sensors and camera viewpoints introduce a considerable gap between training and testing distributions [19].

Earlier works [52, 19, 25, 21, 10, 26] primarily used GAN-based approaches [9] to synthesize photorealistic thermal images, while recent studies [62, 38, 35] explore diffusion-based methods for higher fidelity. However, these works are mostly limited by narrow training datasets, reducing generalization across diverse distributions. DiffV2IR [38] is a notable exception, combining public datasets to train a semantic- or text-guided diffusion model. Our work differs in two key ways: (1) we evaluate across satellite-aerial, aerial, and ground datasets, beyond DiffV2IR's driving-view-focused evaluation; and (2) our style-disentangled model generates target-style thermal images by conditioning on dataset-specific style embeddings, eliminating the need for retraining as required by DiffV2IR.

**Diffusion- and Flow-based General Image Translation.** The success of diffusion-based [15] and flow-based general image generation [31] has enabled advanced image translation methods like Palette [41] for tasks such as colorization and editing. Latent-space models [39, 4] further improved conditional generation quality, leading to a range of high-performance translation techniques. BBDM [28] models domain transitions via a stochastic Brownian Bridge, while Kwon et al. [23] use contrastive learning on DINOv2 [36] features for content-style fusion. Recent style transfer methods [49, 3, 61, 55, 5] also explore text- and style-image-guided transfer. However, RGB-T image translation poses unique challenges due to fundamental modality differences, which prevent direct weight sharing between RGB and thermal domains. Additionally, effective translation requires learning complex cross-modal relationships across diverse RGB-T datasets. These challenges hinder the applicability of RGB style transfer methods for a robust RGB-T image translation model.

## 3 Methodology

### 3.1 Preliminaries

We adopt a flow-based generative paradigm [31] for thermal image synthesis and provide an overview of their underlying process, following the perspective introduced by the Scalable Interpolate Transformer (SiT) [34]. Let $\mathbf{z}_0 \sim p(\mathbf{z})$ denote the latent variable from data, in accordance with the latent diffusion paradigm [39], and let $\boldsymbol{\epsilon} \sim \mathcal{N}(\mathbf{0}, \mathbf{I})$ represent standard Gaussian noise. The diffusion process is formulated as a time-dependent continuous process over $t \in [0, 1]$, given by:

$$\mathbf{z}_t = \alpha_t \mathbf{z}_0 + \sigma_t \boldsymbol{\epsilon}. \tag{1}$$

Here, $\alpha_t$ is a decreasing function with $\alpha_0 = 1$ and $\alpha_1 = 0$, while $\sigma_t$ is a increasing function such that $\sigma_0 = 0$ and $\sigma_1 = 1$. A simple illustrative example of these schedules is $\alpha_t = 1 - t$ and $\sigma_t = t$, with derivatives $\dot{\alpha}_t = -1$ and $\dot{\sigma}_t = 1$. This process can be modeled using a probability flow Ordinary Differential Equation (ODE):

$$\dot{\mathbf{z}}_t = v(\mathbf{z}_t, t), \tag{2}$$

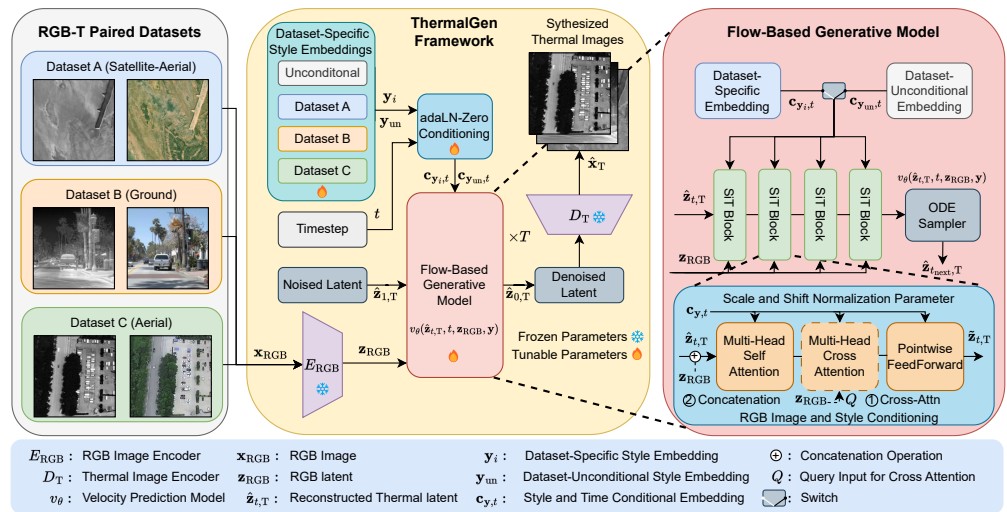

Figure 2: **Overview of ThermalGen**. We sample paired RGB-T data from a diverse collection of satellite-aerial, aerial, and ground datasets for training and evaluation. During thermal image synthesis, the generative model predicts the velocity for $\hat{\mathbf{z}}_{t,\mathrm{T}}$, conditioned on the timestep $t$, the selected dataset-specific style embedding $\mathbf{y}$, and RGB latent $\mathbf{z}_{\mathrm{RGB}}$. After $T$ steps of velocity prediction and denoising, the thermal decoder $D_T$ is used to decode $\hat{\mathbf{z}}_{0,\mathrm{T}}$ and generate the thermal image $\hat{\mathbf{x}}_T$.

where $v$ denotes the velocity function, defined as the expected time derivative of $\mathbf{z}_t$:

$$v(\mathbf{z}_t, t) = \mathbb{E}[\dot{\mathbf{z}}_t \mid \mathbf{z}_t = \mathbf{z}] = \dot{\alpha}_t \mathbb{E}[\mathbf{z}_0 \mid \mathbf{z}_t = \mathbf{z}] + \dot{\sigma}_t \mathbb{E}[\boldsymbol{\epsilon} \mid \mathbf{z}_t = \mathbf{z}]. \tag{3}$$

To approximate this velocity function, we employ a neural network $v_\theta$, parameterized by $\theta$, which is trained by minimizing the following objective:

$$\mathcal{L}_{\mathrm{flow}} = \mathbb{E}_{\mathbf{z}_t, t} \left[ \left\| v_\theta(\mathbf{z}_t, t) - v(\mathbf{z}_t, t) \right\|^2 \right]. \tag{4}$$

Once trained, the model can obtain the denoised latent variable $\hat{\mathbf{z}}_0$ from the terminal noise state $\mathbf{z}_1 = \boldsymbol{\epsilon}$ by integrating the learned reverse flow dynamics governed by $v_\theta$.

## 3.2 RGB-Image-Conditioning and Style-Disentangled Generative Model

The overview of ThermalGen is shown in Fig. 2. Our objective is to synthesize thermal images conditioned on an RGB image and a style embedding disentangled from the model parameters. We define the *RGB-T style* as the mapping relationship between RGB and thermal images, which is influenced by factors such as thermal sensor characteristics, camera viewpoints, and environmental conditions. Let $p(\mathbf{x}_\mathrm{T})$ denote the distribution of a thermal image $\mathbf{x}_\mathrm{T} \in \mathbb{R}^{H \times W \times 1}$. ThermalGen models the conditional distribution $p(\mathbf{x}_\mathrm{T} \mid \mathbf{x}_{\mathrm{RGB}}, \mathbf{y})$, which captures the likelihood of generating $\mathbf{x}_\mathrm{T}$ given an RGB image $\mathbf{x}_{\mathrm{RGB}} \in \mathbb{R}^{H \times W \times 3}$ and a dataset-specific style embedding $\mathbf{y} \in \mathbb{R}^{1 \times D}$. Here, $H$ and $W$ refer to the image height and width, and $D$ denotes the style embedding dimensionality.

**Thermal Image Encoder and Decoder.** To enhance computational efficiency, we adopt the latent diffusion framework [39]. Within this framework, flow and diffusion-based operations are performed on a latent variable $\mathbf{z}_\mathrm{T} = E_\mathrm{T}(\mathbf{x}_\mathrm{T})$, where $E_\mathrm{T}$ is the thermal image encoder. The latent representation $\mathbf{z}_\mathrm{T} \in \mathbb{R}^{\frac{H}{f} \times \frac{W}{f} \times C}$ is a compressed form of the thermal image, with spatial dimensions reduced by a factor of $f$ and $C$ representing the number of latent channels. The synthesized thermal image $\hat{\mathbf{x}}_\mathrm{T}$ is reconstructed from the latent variable $\hat{\mathbf{z}}_\mathrm{T}$, obtained through the flow-based generative model (Section 3.1), using a decoder $D_\mathrm{T}$, such that $\hat{\mathbf{x}}_\mathrm{T} = D_\mathrm{T}(\hat{\mathbf{z}}_\mathrm{T})$.

**Flow-based Latent Generation.** We employ SiT [34] for flow-based latent generation (Section 3.1), leveraging its scalable diffusion transformer architecture [37]. Let $v_\theta(\hat{\mathbf{z}}_{t,\mathrm{T}}, t, \mathbf{z}_{\mathrm{RGB}}, \mathbf{y})$ denote the velocity predicted by the SiT blocks, conditioned on the noised thermal latent $\hat{\mathbf{z}}_{t,\mathrm{T}}$ at timestep $t$, the RGB latent $\mathbf{z}_{\mathrm{RGB}}$, and the style embedding $\mathbf{y}$. Using the ODE sampler, we update the thermal latent to the next timestep, obtaining $\hat{\mathbf{z}}_{t_{\mathrm{next}},\mathrm{T}}$. By iteratively predicting velocities and denoising over $T$ steps, we progressively reconstruct the thermal latent $\hat{\mathbf{z}}_{0,\mathrm{T}}$ from the initially noised latent $\hat{\mathbf{z}}_{1,\mathrm{T}}$.

Table 1: **Overview of RGB-T paired datasets.** Sample numbers that correspond to each dataset's train/validation/test splits are provided. A total of **200k** samples are used for large-scale training. *For the satellite-aerial datasets, the number of samples is computed using a sample stride of $35\text{m}$, indicating the gap between adjacent samples.

| Dataset Name | Environment | Resolution | Sample Number | Splits | Downstream Tasks |
|---|---|---|---|---|---|
| *Satellite-Aerial Datasets* | | | | | |
| Boson-night* [57] | Wild | $512 \times 512$ | $13,011/13,011/26,568$ | train/val/test | Image Alignment |
| DJI-day* (Ours) | Wild | $512 \times 512$ | $19,392$ | train | Image Alignment |
| Bosonplus-day* (Ours) | Wild | $512 \times 512$ | $50,882/4,329$ | train/val | Image Alignment |
| Bosonplus-night* (Ours) | Wild | $512 \times 512$ | $50,695/10,146$ | train/val | Image Alignment |
| *Aerial Datasets* | | | | | |
| CalTech [25] | Wild | $960 \times 600$ | $2,282$ | train | Semantic Segmentation |
| LLVIP [19] | Urban | $1280 \times 1024$ | $12,025/3,463$ | train/test | Human Detection |
| NII-CU [44] | Wild | $2706 \times 1980$ | $2,653/485$ | train/val | Human Detection |
| AVIID [10] | Urban | $464 \times 464$ | $2,412/804$ | train/test | Object Detection |
| *Ground Datasets* | | | | | |
| M$^3$FD [32] | Diverse | Diverse | $3,360/840$ | train/test | Object Detection |
| Freiburg-day [51] | Urban | $1920 \times 650$ | $12,168/32$ | train/test | Semantic Segmentation |
| Freiburg-night [51] | Urban | $1920 \times 650$ | $8,683/32$ | train/test | Semantic Segmentation |
| SMOD-day [2] | Urban | $640 \times 512$ | $5,378$ | train | Object Detection |
| SMOD-night [2] | Urban | $640 \times 512$ | $3,298$ | train | Object Detection |
| MSRS [48] | Urban | $640 \times 480$ | $1,083/361$ | train/test | Image Fusion |
| KAIST [17] | Urban | $640 \times 512$ | $8,643$ | train | Human Detection |
| FLIR [8, 59] | Urban | $640 \times 512$ | $4,129/1,013$ | train/test | Object Detection |

**Style-Disentangled Mechanism.** To address the diverse RGB-T styles from different datasets, we introduce a style-disentangled mechanism that leverages dataset-specific style embeddings for conditional image generation. We define a set of learnable style embeddings $Y = \{\mathbf{y}_0, \mathbf{y}_1, \ldots, \mathbf{y}_n, \mathbf{y}_{\text{un}}\}$, where $n$ denotes the number of datasets or distinct user-defined RGB-T styles, and $\mathbf{y}_{\text{un}}$ represents an unconditional style embedding used for generating thermal images without specifying a style. Given a style embedding $\mathbf{y}_i$ for the $i^{\text{th}}$ style and a timestep $t$, we adopt adaLN-Zero conditioning [37] to generate a corresponding condition embedding $\mathbf{c}_{\mathbf{y}_i, t}$. adaLN-Zero applies adaptive layer normalization, where the scale and shift parameters are modulated based on $\mathbf{y}_i$ and $t$. The selection of conditioning methods is motivated by the findings of AdaIN [16], which demonstrated that altering normalization parameters effectively achieves style transfer by modifying feature statistics. For the unconditional style embedding $\mathbf{y}_{\text{un}}$, we denote the resulting condition embedding as $\mathbf{c}_{\mathbf{y}_{\text{un}}, t}$. During training, the model randomly selects between $\mathbf{c}_{\mathbf{y}_i, t}$ and $\mathbf{c}_{\mathbf{y}_{\text{un}}, t}$ enabling both Classifier-Free Guidance (CFG) [14] and dataset-unconditional generation. This design allows for straightforward extension to additional datasets by appending new style embeddings, thus enhancing adaptability across varied domains.

**RGB Image Conditioning Architecture.** To incorporate RGB image information into the generative model, we utilize a pretrained KL-VAE encoder [39] as the RGB encoder $E_{\text{RGB}}$, which extracts the RGB latent representation $\mathbf{z}_{\text{RGB}}$. We explore two variants for RGB image conditioning. ① *Multi-head Cross-Attention (Cross-Attn)*: we use $\mathbf{z}_{\text{RGB}}$ as the query, while $\hat{\mathbf{z}}_{t,\text{T}}$ serves as both the key and value within an additional cross-attention module embedded in the SiT blocks:

$$\tilde{\mathbf{z}}_{t,\text{T}} = \text{Cross-Attn}(\mathbf{z}_{\text{RGB}}, \hat{\mathbf{z}}_{t,\text{T}}, \hat{\mathbf{z}}_{t,\text{T}}), \tag{5}$$

where $\tilde{\mathbf{z}}_{t,\text{T}}$ represents the intermediate features produced by the attention operation. This module is inserted between the self-attention and pointwise feedforward layers in each SiT block. The query-key-value assignment is inspired by style transfer techniques [3, 5], where the content representation serves as the query and the style representation as the key and value. ② *Concatenation*: Alternatively, we can also concatenate $\mathbf{z}_{\text{RGB}}$ with $\hat{\mathbf{z}}_{t,\text{T}}$ to form the input to the SiT blocks:

$$\bar{\mathbf{z}}_{t,\text{T}} = \text{Concatenate}(\hat{\mathbf{z}}_{t,\text{T}}, \mathbf{z}_{\text{RGB}}), \tag{6}$$

where $\bar{\mathbf{z}}_{t,\text{T}}$ denotes the input to the SiT blocks. This method enables convenient fine-tuning from pretrained SiT weights by directly extending the input representations.

# 4 Experiment Setup

**Datasets.** To train a adaptive RGB-T translation model and ensure robust evaluation, we utilize over ten publicly available RGB-T paired datasets, applying a thorough process of curation, filtering, and preprocessing. This includes standardizing data formats, normalizing thermal data to the 8-bit range,

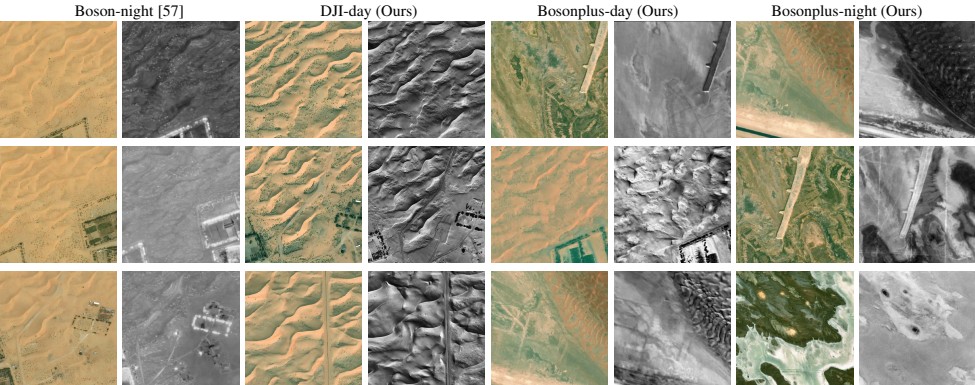

Figure 3: **Visual comparison between the Boson-night dataset [57] and our curated datasets.** Each dataset illustrates differences in thermal sensors, lighting conditions, and geography. Columns show paired satellite RGB and corresponding 8-bit thermal images.

aligning RGB and thermal image pairs, and removing unusable data and regions with invalid thermal readings. Additionally, we introduce three newly collected satellite-aerial RGB-T datasets to enhance translation performance between remote sensing RGB and aerial thermal imagery. A summary of the datasets used is presented in Table 1. The datasets are grouped into three categories: satellite-aerial datasets, aerial datasets, and ground datasets.

*Satellite-Aerial Datasets.* These datasets, such as the Boson-night dataset [57] and our curated datasets—**DJI-day**, **Bosonplus-day**, and **Bosonplus-night** (where the prefix denotes the sensor model and the suffix indicates the time of data acquisition)—comprise paired satellite RGB and aerial thermal images. Details about the data collection process are in Appendix Sec. A. These aligned datasets enable a range of applications, including paired image translation [57], multi-modal image alignment [58, 56], and multi-modal place recognition [57]. A comparison between the Boson-night dataset [57] and our datasets is provided in Appendix Sec. D, highlighting our contributions in terms of broader geographic coverage, increased diversity of thermal sensors, and the inclusion of both daytime and nighttime imagery. Figure 3 offers a visual comparison that illustrates variations across thermal sensors, temporal differences, and the rich geographical diversity captured in our datasets.

*Aerial Datasets.* These datasets include data captured by UAVs or surveillance cameras, featuring oblique aerial views of thermal images. We include four aerial datasets for our training and evaluation: CalTech [25] and NII-CU [44] contain thermal images captured in natural environments, while LLVIP [19] features data from surveillance cameras in the urban or campus environment. Additionally, AVIID [10] contains aerial RGB and thermal image pairs collected via a dual-light camera system in an urban environment. Most datasets were collected in daytime settings, though LLVIP and AVIID notably include nighttime RGB and thermal data.

*Ground Datasets.* These datasets consist of paired imagery from handheld or vehicle-mounted cameras providing horizontal-view thermal data. We incorporate six datasets for training and evaluation: The M$^3$FD dataset [32] delivers RGB-T paired data across varied environments, including urban and wilderness settings, featuring a broad range of objects captured in both daylight and nighttime conditions. The Freiburg [51] and SMOD [2] datasets contribute sequential street-view thermal imagery with day-night change. MSRS [48] provides aligned RGB-T image pairs with high thermal contrast for object detection applications. KAIST Multispectral Pedestrian Detection Benchmark [17] offers regular traffic scenes captured via vehicle-mounted systems for pedestrian detection research. FLIR aligned dataset [59] provides RGB-T paired data from driving scenarios with diverse object classes, including vehicles, pedestrians, and traffic signage.

**Metrics.** We employ four image quality metrics for evaluation: Peak Signal-to-Noise Ratio (PSNR) quantifies image fidelity by measuring the mean-square-error. Structural Similarity Index Measure (SSIM) [54] assesses image structural similarity. Fréchet Inception Distance (FID) [13] measures distribution similarity between predicted and ground truth imagery by comparing 2048-dimensional feature vectors extracted via the Inception v3 model [46]. Learned Perceptual Image Patch Similarity (LPIPS) [60] quantifies perceptual similarity through features from AlexNet [22].

Table 2: **Comparison of RGB-T translation performance on satellite-aerial datasets between baseline methods and ThermalGen.** *Fine-tuned on M$^3$FD; $^+$Fine-tuned on FLIR. The best results are in **bold**, and the second and third best are underlined. This is followed in subsequent tables.

| Methods | Categories | Boson-night | | | | Bosonplus-day | | | | Bosonplus-night | | | |
|---|---|---|---|---|---|---|---|---|---|---|---|---|---|
| | | PSNR↑ | SSIM↑ | FID↓ | LPIPS↓ | PSNR↑ | SSIM↑ | FID↓ | LPIPS↓ | PSNR↑ | SSIM↑ | FID↓ | LPIPS↓ |
| pix2pix [18, 57] | Paired GAN | 23.71 | 0.79 | 149.55 | 0.31 | 14.04 | 0.30 | 170.45 | 0.44 | 19.93 | 0.70 | 137.74 | 0.40 |
| CycleGAN [63, 12] | Unpaired GAN | 17.27 | 0.50 | 119.62 | 0.42 | 12.62 | 0.21 | 279.16 | 0.52 | 11.36 | 0.47 | 105.36 | 0.48 |
| pix2pixHD [53] | Paired GAN | 21.46 | 0.75 | **106.33** | **0.26** | 12.85 | 0.21 | 157.65 | 0.43 | 16.79 | 0.71 | 89.26 | 0.35 |
| VQGAN [7] | Paired GAN | **24.55** | **0.81** | 207.12 | 0.29 | 14.10 | 0.28 | 185.41 | 0.46 | 18.49 | **0.76** | 286.74 | **0.33** |
| DDIM [39, 43] | Paired Diffusion | 18.31 | 0.72 | 203.05 | 0.50 | 12.50 | 0.20 | 261.03 | 0.71 | 15.22 | 0.72 | 112.38 | 0.49 |
| BBDM [28] | Paired Diffusion | 17.85 | 0.62 | 141.27 | 0.37 | 12.42 | 0.18 | 137.68 | 0.46 | 13.88 | 0.62 | 101.08 | 0.43 |
| DiffV2IR [38] | Paired Diffusion | 15.47 | 0.50 | 150.11 | 0.47 | 11.01 | 0.17 | 215.20 | 0.59 | 13.76 | 0.55 | 96.42 | 0.50 |
| DiffV2IR* [38] | Paired Diffusion | 11.69 | 0.18 | 253.82 | 0.66 | 8.83 | 0.11 | 260.42 | 0.52 | 10.01 | 0.25 | 154.76 | 0.66 |
| DiffV2IR$^+$ [38] | Paired Diffusion | 15.55 | 0.46 | 137.74 | 0.49 | 12.32 | 0.19 | 234.54 | 0.56 | 12.01 | 0.52 | **72.02** | 0.48 |
| ThermalGen-L/2 | Paired Diffusion | 21.88 | 0.71 | 161.22 | 0.32 | **14.66** | **0.31** | 76.91 | 0.35 | **20.47** | **0.76** | 75.80 | 0.34 |

Table 3: **Comparison of RGB-T translation performance on aerial datasets between baseline methods and ThermalGen.** *Fine-tuned on M$^3$FD; $^+$Fine-tuned on FLIR; $^†$DiffV2IR models use LLVIP test set for training.

| Methods | Categories | LLVIP | | | | NII-CU | | | | AVIID | | | |
|---|---|---|---|---|---|---|---|---|---|---|---|---|---|
| | | PSNR↑ | SSIM↑ | FID↓ | LPIPS↓ | PSNR↑ | SSIM↑ | FID↓ | LPIPS↓ | PSNR↑ | SSIM↑ | FID↓ | LPIPS↓ |
| pix2pix [18, 57] | Paired GAN | 12.09 | 0.37 | 326.14 | 0.53 | 17.31 | 0.81 | 168.77 | 0.37 | 21.41 | 0.61 | 146.26 | 0.30 |
| CycleGAN [63, 12] | Unpaired GAN | 10.39 | 0.24 | 227.15 | 0.61 | 16.43 | 0.77 | 125.37 | 0.37 | 15.54 | 0.46 | 91.37 | 0.32 |
| pix2pixHD [53] | Paired GAN | 11.51 | 0.33 | 281.89 | 0.51 | 19.46 | 0.80 | 118.60 | 0.32 | 20.01 | 0.56 | 127.63 | 0.27 |
| VQGAN [7] | Paired GAN | 11.75 | 0.36 | 273.06 | 0.58 | 15.53 | 0.81 | 173.37 | 0.37 | 21.71 | 0.60 | 96.46 | 0.23 |
| DDIM [39, 43] | Paired Diffusion | 10.94 | 0.41 | 297.26 | 0.71 | 17.79 | 0.77 | 180.14 | 0.48 | 10.96 | 0.36 | 290.46 | 0.76 |
| BBDM [28] | Paired Diffusion | 9.98 | 0.22 | 313.54 | 0.67 | 14.36 | 0.71 | 118.59 | 0.42 | 18.53 | 0.49 | 141.06 | 0.31 |
| DiffV2IR [38] | Paired Diffusion | **22.17**$^†$ | **0.77**$^†$ | **50.10**$^†$ | **0.11**$^†$ | 12.28 | 0.63 | 159.99 | 0.50 | 18.17 | 0.53 | 51.61 | 0.21 |
| DiffV2IR* [38] | Paired Diffusion | 14.07$^†$ | 0.51$^†$ | 174.79$^†$ | 0.46$^†$ | 14.88 | 0.65 | 135.15 | 0.44 | 14.32 | 0.45 | 116.11 | 0.49 |
| DiffV2IR$^+$ [38] | Paired Diffusion | 10.23$^†$ | 0.40$^†$ | 157.35$^†$ | 0.43$^†$ | 13.76 | 0.67 | 132.00 | 0.44 | 11.00 | 0.38 | 112.97 | 0.49 |
| ThermalGen-L/2 | Paired Diffusion | 11.12 | 0.34 | 238.60 | 0.51 | **26.44** | **0.92** | **69.30** | 0.21 | **24.89** | **0.75** | **29.05** | **0.13** |

**Baselines.** We evaluate ThermalGen against widely adopted GAN-based methods for RGB-T image translation [25, 12, 57, 10], including pix2pix [18], CycleGAN [63], pix2pixHD [53], and VQGAN [7]. The pix2pix framework employs a U-Net [40]-based generator and a patch-based discriminator, optimized with L1 loss. In comparison, pix2pixHD adopts ResNet blocks [11] and leverages LPIPS loss [60] to enhance perceptual quality. CycleGAN utilizes two generators and two discriminators to translate images into the target domain and reconstruct them back into the source domain, allowing training on unpaired data. VQGAN incorporates vector quantization with a discrete codebook to improve perceptual quality. For diffusion-based baselines, we compare against DDIM [43] and DiffV2IR [38], all of which employ the latent diffusion model architecture [39] for thermal image generation. All baseline models are trained on our curated RGB-T dataset collection to evaluate cross-dataset generalization, except for DiffV2IR [38], for which we use pretrained models.

**Implementation Details.** For joint training across multiple datasets, we randomly sample batches from all available training sets. During training, each image is randomly resized and cropped to $256 \times 256$ pixels. For evaluation, images are resized to $256 \times 256$ pixels with denoising steps $T = 50$. The style embedding dimensionality is $1024$. Additional training details are in Appendix Sec. C.

## 5 Results

### 5.1 Cross-dataset RGB-T Image Translation Performance

Tables 2-4 present a comprehensive performance comparison between our proposed ThermalGen framework and state-of-the-art baseline methods across representative satellite-aerial, aerial, and ground RGB-T datasets. The results show that ThermalGen outperforms both GAN- and diffusion-based baselines across all categories. In particular, it achieves superior perceptual quality, as measured by FID and LPIPS, on datasets such as Bosonplus-day, Bosonplus-night, NII-CU, AVIID, M$^3$FD, and MSRS. The performance advantage is especially notable when compared to fine-tuned DiffV2IR models, which often perform well on isolated datasets but struggle to generalize across diverse imaging conditions. ThermalGen's robust adaptability stems from its architectural design and unified training strategy, which effectively captures the complex RGB-T patterns in heterogeneous datasets.

### 5.2 Ablation Studies

Figure 4 presents the results of ablation studies conducted to identify the most effective architectural design and hyperparameter configurations for ThermalGen.

Table 4: **Comparison of RGB-T translation performance on ground datasets between baseline methods and ThermalGen.** *Fine-tuned on M³FD; +Fine-tuned on FLIR.

| Methods | Categories | M³FD | | | | MSRS | | | | FLIR | | | |
|---|---|---|---|---|---|---|---|---|---|---|---|---|---|
| | | PSNR↑ | SSIM↑ | FID↓ | LPIPS↓ | PSNR↑ | SSIM↑ | FID↓ | LPIPS↓ | PSNR↑ | SSIM↑ | FID↓ | LPIPS↓ |
| pix2pix [18, 57] | Paired GAN | 21.10 | 0.72 | 127.62 | 0.34 | 21.78 | 0.68 | 174.81 | 0.37 | 17.13 | 0.54 | 224.11 | 0.44 |
| CycleGAN [63, 12] | Unpaired GAN | 11.28 | 0.42 | 171.37 | 0.56 | 11.42 | 0.35 | 99.14 | 0.53 | 11.15 | 0.32 | 137.97 | 0.49 |
| pix2pixHD [53] | Paired GAN | 19.37 | 0.67 | 112.31 | 0.28 | 18.21 | 0.61 | 121.05 | 0.35 | 15.63 | 0.49 | 164.53 | 0.37 |
| VQGAN [7] | Paired GAN | 21.02 | 0.71 | 79.21 | 0.26 | 22.27 | 0.69 | 106.51 | 0.38 | 16.95 | 0.50 | 141.00 | 0.40 |
| DDIM [39, 43] | Paired Diffusion | 11.15 | 0.45 | 229.24 | 0.70 | 7.35 | 0.21 | 262.93 | 0.79 | 11.24 | 0.39 | 296.81 | 0.71 |
| BBDM [28] | Paired Diffusion | 17.26 | 0.61 | 120.79 | 0.37 | 20.27 | 0.62 | 145.86 | 0.37 | 16.10 | 0.45 | 177.81 | 0.42 |
| DiffV2IR [38] | Paired Diffusion | 12.24 | 0.42 | 75.95 | 0.45 | 15.16 | 0.54 | 61.52 | 0.33 | 20.76 | 0.52 | 38.88 | 0.21 |
| DiffV2IR* [38] | Paired Diffusion | 22.76 | 0.79 | 37.74 | 0.13 | 10.16 | 0.30 | 104.01 | 0.57 | 11.29 | 0.44 | 106.12 | 0.45 |
| DiffV2IR+ [38] | Paired Diffusion | 12.68 | 0.45 | 78.45 | 0.43 | 6.80 | 0.17 | 113.26 | 0.64 | 22.39 | 0.57 | 37.83 | 0.18 |
| ThermalGen-L/2 | Paired Diffusion | 23.73 | 0.81 | 35.82 | 0.14 | 24.38 | 0.76 | 52.31 | 0.21 | 17.10 | 0.52 | 70.09 | 0.33 |

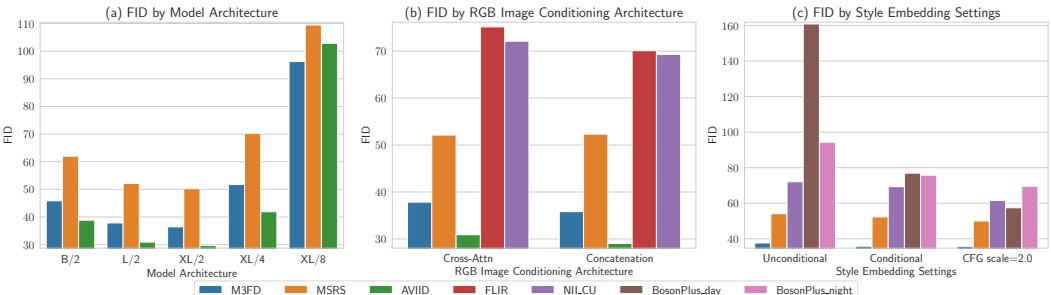

Figure 4: **Ablation studies by comparison of FID scores across different model designs.**

**Transformer and Patch Sizes.** Figure 4(a) summarizes the performance of RGB-T image translation across various transformer sizes (SiT-B, SiT-L, SiT-XL) and patch sizes (2, 4, 8). The results clearly demonstrate that larger transformer sizes (SiT-L and SiT-XL) outperform the smaller variant (SiT-B) in FID scores across all evaluated datasets (M³FD, MSRS, and AVIID). Specifically, SiT-XL/2 achieves the lowest FID scores, indicating superior visual quality and realism. Additionally, models employing smaller patch sizes consistently yield better performance, suggesting that finer granularity for patches substantially enhances the generated image quality.

**RGB Image Conditioning Design.** Figure 4(b) presents an analysis of different configurations for RGB image conditioning, specifically comparing cross-attention and concatenation approaches over five evaluation datasets. The results indicate that concatenating RGB latents yields overall better FID performance than incorporating them as query inputs within the cross-attention module.

**Style Embeddings Settings.** Figure 4(c) evaluates three style embedding configurations: dataset-unconditional style embedding (Unconditional), dataset-specific style embedding (Conditional), and Classifier-Free Guidance (CFG) with scale = 2.0. For datasets with distinctive RGB-T styles (Bosonplus-day, Bosonplus-night, NII-CU), dataset-unconditional style embedding produces degraded FID scores, while CFG (scale = 2.0) outperforms using dataset-specific style embeddings alone. This confirms that style embeddings significantly influence generation quality and encode dataset-specific RGB-T relationships. For general-purpose datasets like M³FD and MSRS, the improvement is marginal, likely because these styles are encoded within the generative model.

## 5.3 Qualitative Results

The visualization results in Fig. 6 demonstrate the superior performance of ThermalGen compared to baseline methods across ground, aerial, and satellite-aerial datasets. GAN-based approaches exhibit notable limitations: Pix2Pix and Pix2PixHD generate distorted thermal imagery, while CycleGAN produces outputs resembling grayscale conversions of RGB images, failing to accurately represent thermal distributions. VQGAN results contain prominent grid artifacts that compromise image quality. For diffusion-based methods, DiffV2IR tends to generate thermal images with excessively sharp boundaries, misrepresenting the gradual transitions in actual heat distributions. In contrast, ThermalGen produces high-fidelity thermal images that accurately match the distribution characteristics of ground truth thermal data. This superior performance persists across diverse thermal sensors, viewpoints, and environmental conditions, effectively bridging the substantial domain gaps between satellite-aerial, aerial, and ground-level datasets. In Fig. 7, we observe that our trained DDIM

Table 5: **FID performance across CFG scale factors.** Bold values indicate optimal settings for each dataset.

| CFG Scale | Boson-night | FLIR |
|---|---|---|
| - | 161.22 | 70.09 |
| 2.0 | 157.57 | 66.54 |
| 4.0 | 126.50 | **63.43** |
| 8.0 | **116.46** | 68.24 |
| 16.0 | 137.95 | – |

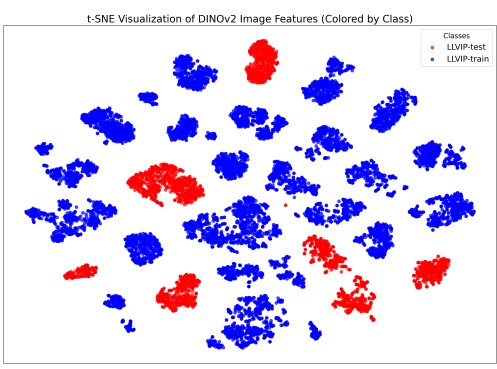

Figure 5: **t-SNE visualization of DINOv2 features for LLVIP thermal images.**

often produces random samples resembling the training distribution rather than being conditioned on the RGB image. This highlights the superior adaptability of ThermalGen compared to DDIM.

### 5.4 Limitations and Discussions

In Tables 2-4, our results indicate performance degradation in specific datasets, including Boson-night, LLVIP, and FLIR datasets. Through root cause analysis of the visualized outputs and systematic investigation, we identify key limitations and propose targeted solutions for each challenging scenario:

- **Boson-night:** The underperformance stems from extremely low-contrast thermal imagery that hampers effective learning, resulting in dark and blurred outputs. However, this limitation can be effectively mitigated by adjusting the classifier-free guidance (CFG) scale factor. Our analysis in Table 5 reveals that varying the CFG scale significantly impacts generation quality, with an optimal CFG factor of 8.0 reducing the FID from 161.22 to 116.46. This finding underscores the effectiveness of our style embedding mechanism, as the CFG scale controls the influence of style embeddings during generation.

- **FLIR:** Performance issues arise from blurred distant objects and extreme lighting conditions (overexposure and underexposure). Similarly, CFG scale adjustment proves effective in Table 5, with optimal performance achieved at CFG factor 4.0, reducing FID from 70.09 to 63.43. While this does not yet surpass state-of-the-art benchmarks, the consistent trend highlights the potential of our framework to adapt under challenging conditions.

- **LLVIP:** The primary challenge is limited scene diversity with predominantly static backgrounds and minimal dynamic content, creating a distribution shift between training and testing conditions. Through t-SNE analysis of DINOv2 features in Fig. 5, we reveal a clear distribution gap between training and testing thermal images, primarily attributed to camera differences. The most effective solution is expanding the training dataset to bridge this distribution gap and improve generalization.

Overall, these results show that while dataset-specific challenges remain, CFG scale adjustment consistently improves performance and supports the effectiveness of our style-disentangled approach, suggesting promising avenues for further enhancement of cross-domain generalization.

## 6 Conclusions

In summary, we presented **ThermalGen**, an adaptive RGB-Thermal (RGB-T) image translation model built on a flow-based generative architecture with RGB image conditioning and a style-disentangling mechanism. Through extensive experiments on a diverse suite of RGB-T datasets—including three newly introduced satellite-aerial datasets—ThermalGen demonstrates strong cross-dataset generalization across different thermal sensors, viewpoints, and environmental conditions. Our analysis highlights the significance of disentangled style embeddings in enhancing translation robustness.

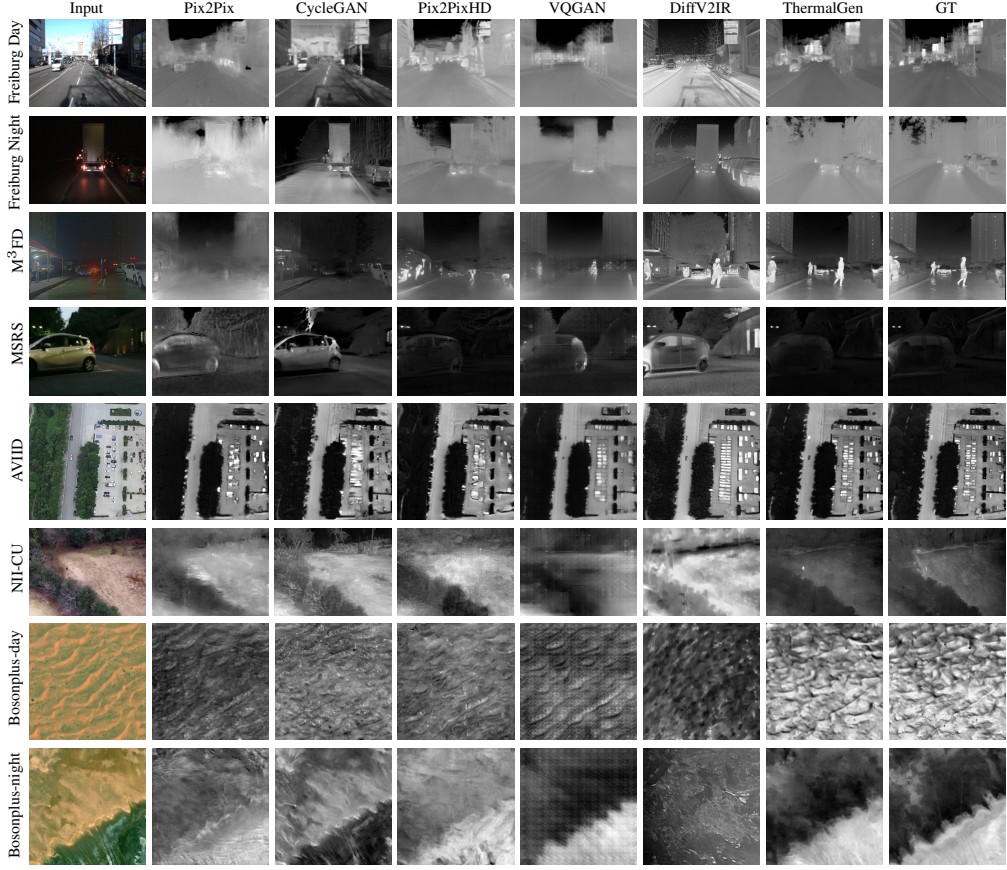

Figure 6: **Visual Comparison of Baseline Methods and ThermalGen**. A comprehensive evaluation is conducted across diverse RGB-T datasets, including ground (Freiburg, M³FD, MSRS), aerial (AVIID, NII-CU), and satellite-aerial (Bosonplus-day, Bosonplus-night) domains, demonstrating the method's adaptability and performance.

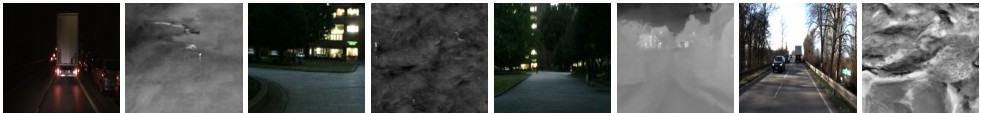

Figure 7: **Failure cases of DDIM.** We show paired RGB input and DDIM output. DDIM tends to generate random samples that align with training data, rather than reflecting the RGB input.

Future work will focus on systematically incorporating synthesized thermal data from multiple sources to further advance cross-modal model training.

**Broader Impacts.** ThermalGen generates realistic thermal images from RGB inputs, facilitating the validation of thermal data realism. However, it also presents potential risks of misuse—especially in safety-critical contexts—where synthetic outputs might be misinterpreted as genuine thermal data.

## Acknowledgments

This work was supported by the Technology Innovation Institute, the NSF CAREER Award 2145277, the NSF CPS Grant CNS-2121391, and the NYU IT High Performance Computing resources, services, and staff expertise. We thank the support from Technology Innovation Institute - ARRC - ViCon team. Giuseppe Loianno serves as consultant for the Technology Innovation Institute. This arrangement has been reviewed and approved by the New York University in accordance with its policy on objectivity in research.

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

## A  Satellite-Aerial Dataset Collection Process

Our data collection process involves the following steps: (1) deploying UAVs over selected areas to capture thermal image patches embedded with GPS metadata; (2) generating a unified thermal orthomosaic[1]; (3) cropping and aligning corresponding regions from satellite imagery with a spatial resolution of $1\text{m/pixel}$; (4) excluding regions with invalid thermal data; and (5) applying grid sampling to extract square image patches for model training and evaluation (following the setup in [57], we adopt a sampling stride of $35\text{m}$ and a crop size of $512 \times 512$). This process ensures the precise alignment between satellite RGB and thermal images.

## B  Dataset Preprocessing Details

For large-scale training, all datasets are converted into the WebDataset[2] format, except for the satellite-aerial datasets. These datasets consist of paired, spatially aligned thermal and satellite RGB maps, from which we randomly sample regions across the entire map. We also detail the specific preprocessing steps applied to each dataset to ensure reproducibility:

- **NII-CU:** Resized RGB images to match thermal image dimensions.
- **TARDAL:** Created an 80%/20% train/test split by random sampling since no official split was provided in the dataset.
- **Freiburg:** Removed the black padding from thermal images on both right and left sizes.

## C  Additional Training Details

We conduct all training and evaluation using a single NVIDIA A100 or H100 GPU. For training the thermal encoder and decoder, we employ a batch size of 16 and use the AdamW [33] optimizer with a learning rate of $6 \times 10^{-5}$ and a weight decay of $1 \times 10^{-3}$, over a total of 200k training steps. All other configurations follow the default settings of the Latent Diffusion Model [39]. In training the flow-based generative models, we use a batch size of 64 with AdamW optimizer at a learning rate of $1 \times 10^{-4}$ and no weight decay, for 200k training steps.

## D  Dataset Parameters Comparison

A comparison between the Boson-night dataset [57] and our datasets is provided in Table 6, highlighting our contributions in terms of broader geographic coverage, increased diversity of thermal sensors, and the inclusion of both daytime and nighttime imagery.

Table 6: **Comparison of dataset parameters between Boson-night [57] and our datasets.** The differences are highlighted regarding area coverage, satellite map sources, thermal sensors used, collection years, and the number of surveyed regions.

| Dataset Name | Area | Satellite Map | Thermal Sensor | Collection Year | Number of Regions |
|---|---|---|---|---|---|
| Boson-night [57] | $33\text{km}^2$ | Bing | Boson | 2021 | 1 |
| DJI-day (Ours) | $29\text{km}^2$ | ESRI[3] | DJI Zenmuse H20T | 2023 | 1 |
| Bosonplus-day (Ours) | $85\text{km}^2$ | ESRI[3] | Bosonplus | 2024 | 3[4] |
| Bosonplus-night (Ours) | $94\text{km}^2$ | ESRI[3] | Bosonplus | 2024 | 3[4] |

## E  Thermal Auto-encoder Training Details and Reconstruction Performance

This section details the training procedure and reconstruction performance of the thermal encoder $E_\text{T}$ and decoder $D_\text{T}$. Both components are trained using the KL-VAE framework [39] on our curated

---

[1] We used Agisoft Metashape to generate the orthomosaic: https://www.agisoft.com/

[2] https://github.com/webdataset/webdataset

[3] ESRI satellite map: https://www.esri.com/en-us/home

[4] For compliance, we will release data only for 2 regions in Bosonplus-day and Bosonplus-night.

Table 7: **Reconstruction FID and PSNR performance across multiple RGB-T datasets**

| Method | FLIR | LLVIP | AVIID | MSRS | NII-CU | M³FD | Bosonplus-day | Bosonplus-night | Boson-night |
|---|---|---|---|---|---|---|---|---|---|
| | | | | | **FID ↓** | | | | |
| klvae w/o GAN loss | 18.51 | 2.94 | 18.16 | 14.37 | 9.07 | 5.50 | 16.63 | 6.48 | 4.24 |
| klvae w/ GAN loss | 14.66 | 3.14 | 12.21 | 12.20 | 8.67 | 5.33 | 6.73 | 3.52 | 2.00 |
| | | | | | **PSNR ↑** | | | | |
| klvae w/o GAN loss | 30.10 | 37.63 | 31.29 | 38.73 | 45.73 | 40.12 | 30.46 | 41.32 | 43.53 |
| klvae w/ GAN loss | 28.74 | 36.59 | 30.07 | 37.60 | 45.24 | 39.03 | 29.15 | 40.84 | 43.16 |

Table 8: **Style vs. CLIP Embeddings in ThermalGen.**

| Method | Bosonplus-day ↓ | Bosonplus-night ↓ | FLIR ↓ |
|---|---|---|---|
| ThermalGen (Style embedding) | 76.91 | 75.80 | 70.09 |
| ThermalGen (CLIP embedding) | 106.08 | 89.65 | 75.49 |

RGB-T dataset collection, optimized with a combination of reconstruction losses (L1 and LPIPS [60]) and KL regularization on the latent space. After 300 epochs (200k steps) of training, we optionally fine-tune the final decoder layers for 75 epochs using a GAN loss plus the loss mentioned above, as in the original LDM [39], to enhance FID scores without altering the latent representation, but it will slightly negatively affect PSNR metrics. We note that the results reported in the paper use the decoder **without** GAN loss.

## F  Comparison of Style and CLIP Embeddings

We compare ThermalGen with two embedding strategies: (1) fixed style embeddings and (2) CLIP-based embeddings extracted from paired RGB images. As shown in Table 8, fixed style embeddings yield substantially lower FID scores on the Bosonplus-day and Bosonplus-night datasets, while achieving comparable performance on FLIR. Moreover, using fixed style embeddings improves computational efficiency by removing the need to compute CLIP features, demonstrating both the effectiveness and efficiency of our approach.

## G  Evaluation Dataset Comparison

We provide a detailed comparison of the evaluation datasets in Table 9, organized by key sources of variation. To further illustrate domain differences, we present t-SNE visualizations of thermal features in Fig. 8 extracted using DINOv2 for the following dataset pairs: *Bosonplus-night vs. Boson-night* (sensor variation), *Bosonplus-day vs. AVIID* (viewpoint variation), and *Bosonplus-day vs. Bosonplus-night* (diurnal variation). These analyses reveal significant feature discrepancies across conditions, while demonstrating that our model effectively adapts to such shifts.

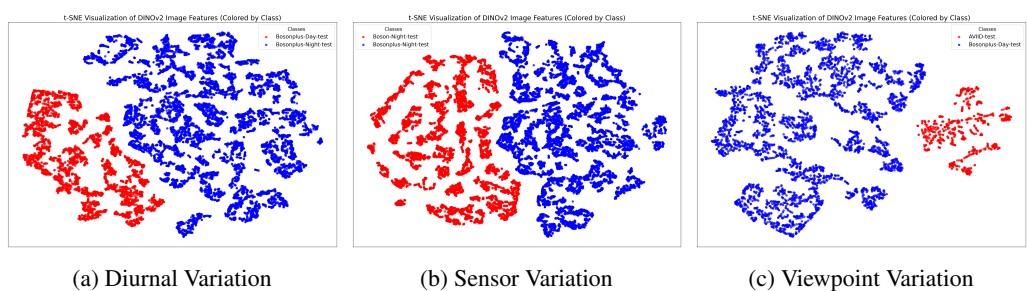

(a) Diurnal Variation      (b) Sensor Variation      (c) Viewpoint Variation

Figure 8: **t-SNE Analysis for Diurnal, Sensor, and Viewpoint Variation of Datasets**

Table 9: **Evaluation Dataset Attribute Comparison.** Attributes include diurnal setting, thermal sensor configuration, viewpoint, and environment type.

| Dataset | Diurnal | Sensor Configuration | Viewpoint | Environment |
|---|---|---|---|---|
| Boson-night | Night | FLIR Boson | Satellite–aerial | Wild |
| Bosonplus-day | Day | FLIR Bosonplus | Satellite–aerial | Wild |
| Bosonplus-night | Night | FLIR Bosonplus | Satellite–aerial | Wild |
| LLVIP | Day/Night | Vanadium Oxide Uncooled Focal Plane Arrays | Aerial | Urban |
| NII-CU | Day | FLIR Vue Pro | Aerial | Wild |
| AVIID | Day/Night | Infrared ($640\times480$, 8–14 $\mu$m) | Aerial | Urban |
| M3FD | Day/Night | Infrared ($640\times512$, 8–14 $\mu$m) | Ground | Diverse |
| MSRS | Day/Night | InfRec R500 | Ground | Urban |
| FLIR | Day/Night | FLIR Tau 2 | Ground | Urban |

