# OpenReview forum: "ThermalGen: Style-Disentangled Flow-Based Generative Models for RGB-to-Thermal Image Translation"
_NeurIPS.cc/2025/Conference — NeurIPS 2025 poster_

### Official Review · Reviewer_N5UX · 2025-06-08

**Clarity:** 4
**Significance:** 3
**Originality:** 4
**Rating:** 5
**Confidence:** 5

**Summary:**

The authors introduce ThermalGen, a flow-based generative model designed for RGB-to-Thermal (RGB-T) image translation. ThermalGen addresses the scarcity of paired RGB-T data, which is crucial for multi-modal tasks, by synthesizing thermal images from readily available RGB inputs. The model incorporates an RGB image conditioning architecture and a style-disentangled mechanism for generating thermal images that reflect significant variations in viewpoints, sensor characteristics, and environmental conditions. In other words, they treat the unique relationship between a collection of a specific kind of imagery and its thermal counterpart as a "style" (in the sense of style transfer), and disentangle this relationship so that their model can adaptively generate thermal imagery based on the kind of RGB input. To support large-scale training, the authors curated eight public RGB-T datasets and introduced three new large-scale satellite-aerial RGB-T datasets: DJI-day, BosonPlus-day, and BosonPlus-night. Their evaluations demonstrate that ThermalGen achieves comparable or superior translation performance compared to existing GAN-based and diffusion-based methods. The authors state that their model is the first with such broad RGB-T synthesis capabilities.

On the specifics of their evaluation, the evaluation of ThermalGen involved experiments across various RGB-T benchmarks, including satellite-aerial, aerial, and ground datasets. The performance was assessed using four image quality metrics: Peak Signal-to-Noise Ratio (PSNR), Structural Similarity Index Measure (SSIM), Fréchet Inception Distance (FID), and Learned Perceptual Image Patch Similarity (LPIPS). ThermalGen was compared against widely adopted GAN-based methods (pix2pix, CycleGAN, pix2pixHD, VQGAN) and diffusion-based methods (DDIM and DiffV2IR). Ablation studies were conducted to determine optimal architectural designs and hyperparameter configurations, examining different transformer and patch sizes, RGB image conditioning designs (cross-attention versus concatenation), and various style embedding settings. Qualitative results were also presented to visually compare ThermalGen's output with baselines across diverse datasets and conditions.

**Questions:**

I don't have any questions since this paper was a delight to read and felt very wholly packaged in terms of problem chosen, motivation, experiments performed, and results.

**Ethical Concerns:**

["NO or VERY MINOR ethics concerns only"]

**Final Justification:**

Nothing's changed in my opinion of the paper and I had no questions I could come up with. The authors have done a great job, in my opinion.

**Limitations:**

Yes.

**Quality:**

4

**Strengths And Weaknesses:**

Strengths:
- Quality:
    - Technically sound submission with well supported claims via an extensive evaluation across numerous benchmarks and competing baselines. Specifically, nine datasets, four metrics, and eight competing baselines, covering GAN-based methods and diffusion-based methods.
    - The authors are honest in evaluating both the strengths and weaknesses of their work. For example, they point out ThermalGen's limitations when dealing with settings with homogeneous samples, low-contrast thermal features, or extreme lighting conditions.
- Clarity:
    - The language is clear, the narrative flows smoothly and sensibly, and the quantitative and qualitative results are presented very clearly. It was an incredibly easy and smooth read and I feel confident that I can replicate their results from reading the main paper and appendix.
- Significance:
    - Although the quantitative results didn't blow me away (their model is the best on average, but it wasn't a clean sweep), it wasn't until I saw the qualitative results when I was really impressed. Specifically, Figure 5. There is a very clear difference between the quality of ThermalGen's outputs and the competing models'. It's quite significant and I believe this alone would convince researchers working in this space to build off of this model and treat it as the new state of the art.
- Originality:
    - Although generative modelling of RGB and thermal image pairs has been done many times, this is the first paper to use a flow-based generative model, as far as I know. The authors also introduce their own unique modifications to improve the modelling to make the model adaptive to other settings via the style embedding.
    - The authors introduce three new large-scale datasets, which is almost always a boon for the research community. Curating high quality and large scale datasets is difficult work and I commend the authors for doing so, even if it wasn't truly necessary for motivating their model.
    - Related work has been adequately cited and the authors has made it clear how their work differs from previous contributions.

Weaknesses: No significant weaknesses worth mentioning that I can think of.

---

> ### Author Rebuttal · Authors · 2025-07-29
>
> Thank you for the encouraging and thoughtful review. We appreciate your recognition of our technical contributions, including the flow-based architecture, style-disentangled mechanism, and the curated RGB-T datasets. We're especially grateful for your positive assessment of ThermalGen's quality, clarity, significance, and originality. While the model may not significantly outperform all baselines on every metric, its consistent performance across most datasets and strong qualitative results highlight its robustness and potential. Your feedback affirms our direction and inspires further development of ThermalGen.

---

> > ### Comment · Reviewer_N5UX · 2025-08-03
> >
> > Great work y'all, it was my pleasure reading this paper. Keep it up :)

---

### Official Review · Reviewer_sJDP · 2025-06-18

**Clarity:** 3
**Significance:** 4
**Originality:** 3
**Rating:** 5
**Confidence:** 5

**Summary:**

This paper presents ThermalGen, a flow-based model for RGB-to-thermal image translation. The method introduces an RGB conditioning architecture and style-disentangled mechanism, supported by newly curated datasets including three large-scale satellite-aerial RGB-T pairs.

**Questions:**

- Does the model demonstrate sufficient generalization? Could you validate its effectiveness on thermal-to-RGB tasks as well? Relevant references include:

[1] T2V-DDPM: Thermal to Visible Face Translation using Denoising Diffusion Probabilistic Models

[2] DiffTV: Identity-Preserved Thermal-to-Visible Face Translation via Feature Alignment and Dual-Stage Conditions

**Ethical Concerns:**

["NO or VERY MINOR ethics concerns only"]

**Final Justification:**

I will keep my score.

**Limitations:**

The authors have adequately illustrated limitations in the main text.

**Quality:**

3

**Strengths And Weaknesses:**

Strengths:
- The contribution of three high-quality RGB-thermal datasets is particularly valuable given the scarcity of such data in this field. The planned open-source release would greatly benefit the research community.
- Experimental results demonstrate clear advantages of ThermalGen over competing methods.
- To my knowledge, this represents a comprehensive flow-based approach for RGB-thermal translation, presented in a three-in-one framework. Both novelty and technical contributions meet the standard for NIPS acceptance.

Weaknesses:
- The comparison tables include too few diffusion-based methods (which appear limited in scope). More comprehensive comparisons with general image-to-image approaches (e.g., BBDM) would better demonstrate the method's generalization capability.

---

> ### Author Rebuttal · Authors · 2025-07-29
>
> **Q1: More comprehensive comparisons with general image-to-image approaches (e.g., BBDM) would better demonstrate the method's generalization capability.**
>
>
> Thank you for this suggestion. We train the latent BBDM-f8 with our datasets and evaluate the performance and we attach the result table. The quantitative results show that ThermalGen has consistently better performance than BBDM. We also verify the visualization result showing that trained BBDM . We will add these results to our Table 1-3.
>
> | Methods         | Categories        | Boson-night PSNR↑ | SSIM↑ | FID↓   | LPIPS↓ |   | BosonPlus-day PSNR↑ | SSIM↑ | FID↓   | LPIPS↓ |   | BosonPlus-night PSNR↑ | SSIM↑ | FID↓   | LPIPS↓ |
> |----------------|------------------|-------------------|-------|--------|--------|---|----------------------|-------|--------|--------|---|------------------------|-------|--------|--------|
> |BBDM | Paired Diffusion | 17.26 | 0.61 | 120.79 | 0.37 | | 20.27 | 0.62 | 145.86 | 0.37 | | 16.10 | 0.45 | 177.81 | 0.42
> | ThermalGen-L/2 | Paired Diffusion | 21.88           | 0.71  | 161.22 | 0.32   |   | 14.66            | 0.31 | 76.91 | 0.35 |   | 20.47             | 0.76 | 75.80 | 0.34 |
>
> | Methods         | Categories        | LLVIP PSNR↑ | SSIM↑ | FID↓   | LPIPS↓ |   | NII-CU PSNR↑ | SSIM↑ | FID↓   | LPIPS↓ |   | AVIID PSNR↑ | SSIM↑ | FID↓   | LPIPS↓ |
> |----------------|------------------|-------------|-------|--------|--------|---|---------------|-------|--------|--------|---|--------------|-------|--------|--------|
> | BBDM | Paired Diffusion | 9.98 | 0.22 | 313.54 | 0.67 | | 14.36 | 0.71 | 118.59 | 0.42 | | 18.53 | 0.49 | 141.06 | 0.31 |
> | ThermalGen-L/2 | Paired Diffusion | 11.12       | 0.34  | 238.60 | 0.51   |   | 26.44     | 0.92 | 69.30 | **0.21** |   | 24.89     | 0.75 | 29.05 | 0.13 |
>
> | Methods         | Categories        | M³FD PSNR↑ | SSIM↑ | FID↓   | LPIPS↓ |   | MSRS PSNR↑ | SSIM↑ | FID↓   | LPIPS↓ |   | FLIR PSNR↑ | SSIM↑ | FID↓   | LPIPS↓ |
> |----------------|------------------|------------|-------|--------|--------|---|-------------|-------|--------|--------|---|-------------|-------|--------|--------|
> | BBDM | Paired Diffusion | 17.26 | 0.61 | 120.79 | 0.37 | | 20.27 | 0.62 | 145.86 | 0.37 | | 16.10 | 0.45 | 177.81 | 0.42 |
> | ThermalGen-L/2 | Paired Diffusion | 23.73  | 0.81 | 35.82 | 0.14 |   | 24.38   | 0.76 | 52.31 | 0.21 |   | 17.10      | 0.52 | 70.09 | 0.33 |
>
>
> **Q2: Does the model demonstrate sufficient generalization? Could you validate its effectiveness on thermal-to-RGB tasks as well?**
>
> Thank you for the suggestion. We respectfully note that the Thermal-to-RGB task differs in nature from the RGB-to-Thermal direction and may involve distinct challenges. As such, its effectiveness does not directly reflect the generalization capabilities of our model within RGB-Thermal tasks. Nonetheless, the underlying principles remain consistent: our model employs separate encoders for thermal and RGB modalities, and the flow-based generative model is trained within the latent space of the thermal encoder. A thermal-to-RGB model will instead require to train the flow-based model to generate from the latent distribution from the RGB encoder.

---

> > ### Comment · Reviewer_sJDP · 2025-08-01
> >
> > OK, nice work! I will keep my score. Thank you for your answer.

---

### Official Review · Reviewer_ZBig · 2025-07-02

**Clarity:** 3
**Significance:** 2
**Originality:** 3
**Rating:** 4
**Confidence:** 5

**Summary:**

This paper proposes a flow-based generative model for RGB-to-thermal (RGB-T) image translation, addressing the scarcity of paired RGB-thermal data. Specifically,  first, it introduces an Adaptive flow-based architecture with RGB conditioning and style-disentangled embeddings to handle variations in sensors, viewpoints, and environmental conditions.
Second, it proposes Three new satellite-aerial datasets (DJI-day, BosonPlus-day, BosonPlus-night) for large-scale training. Third, it achieves Superior performance over GAN/diffusion baselines (e.g., pix2pix, CycleGAN, DiffV2IR) across satellite, aerial, and ground benchmarks, validated via PSNR, SSIM, FID, and LPIPS metrics.

**Questions:**

1. Why use dataset-level embeddings instead of learning styles directly from image content (e.g., via attention)? Could this limit adaptation to unseen styles?
2. The motivation of this paper is unconvincing. Why need to translate RGB images into thermal images? They are within different modalities and different visual details. For example, thermal images may contain more texture details of objects than RGB images. If the proposed model works, it is better to show such differences.
3. Thermal images follow physical laws (e.g., heat dissipation). Could incorporating physics-informed losses (e.g., PDE-based regularization) improve accuracy?
4. Flow-based sampling (50 denoising steps) is slower than GANs. Can distillation or quantization achieve real-time translation?
5. Datasets focus on urban/wilderness—how would ThermalGen perform in industrial/medical thermal domains?

**Ethical Concerns:**

["NO or VERY MINOR ethics concerns only"]

**Final Justification:**

After carefully evaluating the motivation, contribution, and response provided by the authors, I decided to increase my rating to 4.

Thanks to the authors' informative response and AC's organization.

**Limitations:**

1. The motivation of this paper is unconvincing. Why need to translate RGB images into thermal images? They are within different modalities and different visual details. For example, thermal images may contain more texture details of objects than RGB images. If the proposed model works, it is better to show such differences.
2. This paper shows significant performance drops in some extreme conditions, for example, blurred distant objects (FLIR dataset) and low-contrast thermal images (Boson-night).
3. The Fixed per-dataset embeddings of the style embedding may not capture intra-dataset variations. However, the authors do not investigate this issue.

**Quality:**

3

**Strengths And Weaknesses:**

Strengths:
1. The flow-based model for RGB-T translation with style disentanglement is reasonable and effective for this image translation task.
2. Unified training on 200k+ diverse samples across 13 datasets shows the potential scalability of this model.
3. Robust performance across domains (satellite, aerial, ground) without retraining.
4. Three new datasets are proposed to contribute to the community.

Weaknesses:
1. The motivation of this paper is unconvincing. Why need to translate RGB images into thermal images? They are within different modalities and different visual details. For example, thermal images may contain more texture details of objects than RGB images. If the proposed model works, it is better to show such differences.
2. This paper shows significant performance drops in some extreme conditions, for example, blurred distant objects (FLIR dataset) and low-contrast thermal images (Boson-night).
3. The Fixed per-dataset embeddings of the style embedding may not capture intra-dataset variations. However, the authors do not investigate this issue.

---

> ### Author Rebuttal · Authors · 2025-07-29
>
> **Q1: The motivation of this paper is unconvincing. Why need to translate RGB images into thermal images? They are within different modalities and different visual details.**
>
> We acknowledge that RGB and thermal images belong to different modalities and capture distinct visual characteristics. However, translating RGB images into thermal images is highly relevant for advancing multimodal computer vision tasks that depend on cross-modality alignment.
>
> Tasks such as RGB-thermal sensor fusion [1], cross-modality image alignment [2], and semantic segmentation [3] frequently require well-aligned RGB-thermal image pairs for supervised training. In practice, acquiring such paired data is challenging due to the need for precise synchronization between RGB and thermal sensors, and the resulting datasets are often limited in scale, diversity, or quality.
>
> Our work aims to address this data scarcity by synthesizing realistic thermal data from RGB inputs, thereby enabling the creation of large-scale pseudo-paired RGB-thermal datasets. These generated pairs can significantly benefit downstream multimodal tasks by enhancing model robustness and reducing dependence on costly real-world data collection. Our approach provides an efficient means for rapid prototyping and proof-of-concept development in cross-modality research. A similar motivation supports recent efforts such as Diffv2IR [6].
>
> Furthermore, we observe the potential of our model to generate multi-pattern plausible thermal outputs conditioned on RGB context by randomizing latent noises inherent in flow-based methods, rather than performing one-to-one translation. This variability helps simulate different possible real-world conditions and improves generalization in downstream applications.
>
> **Q2: This paper shows significant performance drops in some extreme conditions, for example, blurred distant objects (FLIR dataset) and low-contrast thermal images (Boson-night).**
>
> Boson-night: The underperformance on Boson-night can be mitigated by adjusting the classifier-free guidance (CFG) scale factor. The table below illustrates how varying the CFG scale significantly impacts FID performance on this dataset. Notably, a CFG factor of 8.0 yields the best result, reducing the FID to 116.46, which represents a substantial improvement over the default setting.
>
> This finding also underscores the effectiveness of our style embedding mechanism. Since the CFG scale controls the influence of the style embedding during generation, these results demonstrate that appropriately tuning this factor enables the model to generate clearer, higher-fidelity thermal images. Visually, we observe that ThermalGen produces more detailed and realistic outputs at the optimal CFG.
>
> | Method          | Metric       |  CFG Factor = 1.0 (None) | 2.0   | 4.0   | 8.0   | 16.0  |
> |----------------|-------------------|---------|-------|-------|-------|-------|
> | ThermalGen-L/2 | FID ↓             | 161.22  | 157.57| 126.50| **116.46** | 137.95 |
>
> FLIR: Similar to the Boson-night dataset, we observe that the underperformance on FLIR can be alleviated by adjusting the classifier-free guidance (CFG) scale factor. As shown in the table below, tuning the CFG scale leads to a noticeable reduction in FID, with the best performance achieved at a CFG factor of 4.0.
>
> While the resulting FID does not yet surpass SOTA benchmarks, the trend highlights the potential of our style-disentangled framework to adapt under challenging conditions. The visual results further support this observation, showing improved generation quality as the influence of the style embedding is modulated. These findings suggest a promising direction for enhancing cross-domain generalization in extreme or low-quality thermal domains.
>
> | Method          | Metric | CFG Factor = 1.0 (None) | 2.0   | 4.0       | 8.0   |
> |----------------|--------|------------|-------|-----------|-------|
> | ThermalGen-L/2 | FID ↓  | 70.09      | 66.54 | **63.43** | 68.24 |
>
>
> LLVIP: We conduct t-SNE analysis on DINOv2 features for thermal images and reveal a clear distribution gap between training and testing data, primarily due to camera differences. Expanding the training set is the most effective solution. This analysis will be added to the supplementary material.
>
> **Q3: The Fixed per-dataset embeddings of the style embedding may not capture intra-dataset variations.**
>
> We agree that intra-dataset variation can influence performance; however, we argue that such variability should primarily be addressed by the flow-based generative model rather than through the fixed style embedding. For instance, the M3FD dataset contains both daytime and nighttime scenes, yet our model maintains strong performance across this diversity, suggesting its capability to adapt to intra-dataset changes. The style embedding is designed to capture stable, dataset-wide attributes—such as camera specifications and domain characteristics—while the generative model accounts for intra-dataset variation.
>
> **Q4: Why use dataset-level embeddings instead of learning styles directly from image content (e.g., via attention)? Could this limit adaptation to unseen styles?**
>
> The use of dataset-level embeddings in our framework is motivated by the need to encode style information that may not be inferable solely from the RGB image content. For example, consider the satellite-aerial datasets Bosonplus-day and Bosonplus-night. Although their satellite RGB inputs appear visually the same, the corresponding thermal outputs differ significantly due to day/night variation—a distinction that cannot be reliably inferred from RGB appearance alone. In such cases, image-based mechanisms may fail to capture this latent information.
>
> Dataset-level style embeddings also allow the model to incorporate prior knowledge about dataset-specific characteristics. This is particularly beneficial for datasets like NII-CU, which exhibit unique thermal features. As shown in our experiments (Figure 4c), adjusting the classifier-free guidance (CFG) scale—thereby modulating the influence of the style embedding—can lead to notable performance improvements on such datasets. Furthermore, for other datasets like M3FD and MSRS, the inclusion of style embeddings consistently enhances performance, demonstrating that this mechanism improves performance without introducing degradation across diverse thermal domains.
>
> For unseen datasets, our model utilizes an unconditional style embedding that captures a generalized thermal appearance. Alternatively, users may select an existing style embedding that closely matches the target data. For significantly different thermal data, fine-tuning on the specific dataset is recommended.
>
> **Q5: Could incorporating physics-informed losses (e.g., PDE-based regularization) improve accuracy?**
>
> We appreciate this insightful suggestion. Indeed, physics-informed losses have the potential to enhance accuracy, particularly in scenarios involving sequential data—for example, by enforcing consistency based on principles like heat dissipation. However, our current work focuses on single-image translation, where such temporal regularization is less directly applicable. While methods such as PID loss [4] aim to predict physical quantities such as temperature, emissivity, and thermal texture, they typically require fine-tuning the underlying physical model to specific datasets, particularly to account for variations in material thermal properties and sensor configurations. Physics-informed loss represents a valuable direction for future work to enhance the adaptability of ThermalGen.
>
> **Q6: Can distillation or quantization achieve real-time translation?**
>
> We agree that model distillation and quantization are promising strategies for enabling real-time translation. Specifically, model distillation can effectively reduce model complexity and fine-tune on specific datasets. Moreover, recent developments—such as MeanFlow [5]—highlight the feasibility of highly efficient inference through few-step denoising (e.g., 1–2 steps), significantly enhancing speed without compromising quality. These advances point to a promising path toward achieving real-time performance.
>
> **Q7: How would ThermalGen perform in industrial/medical thermal domains?**
>
> We acknowledge the relevance and importance of extending this work to industrial and medical thermal domains. These domains, however, present distinct challenges. Publicly available datasets are often limited in scale and diversity and tend to be object-centric rather than our scene-based datasets, so that the performance in these domains may be suboptimal without adaptation. Nevertheless, ThermalGen provides a strong foundation through its pretrained weights, which capture diverse thermal characteristics across a range of conditions, and will be open-sourced for further adaptation and finetune.
>
> References:
> [1] Brenner, M., Reyes, N. H., Susnjak, T., & Barczak, A. L. (2023). RGB-D and thermal sensor fusion: A systematic literature review. IEEE Access, 11, 82410-82442.
> [2] Ren, J., Jiang, X., Li, Z., Liang, D., Zhou, X., & Bai, X. (2025). Minima: Modality invariant image matching. In Proceedings of the Computer Vision and Pattern Recognition Conference (pp. 23059-23068).
> [3] Shin, U., Lee, K., Kweon, I. S., & Oh, J. (2024, May). Complementary random masking for rgb-thermal semantic segmentation. In 2024 IEEE International Conference on Robotics and Automation (ICRA) (pp. 11110-11117). IEEE.
> [4] Mao, F., Mei, J., Lu, S., Liu, F., Chen, L., Zhao, F., & Hu, Y. (2025). PID: physics-informed diffusion model for infrared image generation. Pattern Recognition, 111816.
> [5] Geng, Z., Deng, M., Bai, X., Kolter, J. Z., & He, K. (2025). Mean flows for one-step generative modeling. arXiv preprint arXiv:2505.13447.
> [6] Ran, L., Wang, L., Wang, G., Wang, P., & Zhang, Y. (2025). Diffv2ir: visible-to-infrared diffusion model via vision-language understanding. arXiv preprint arXiv:2503.19012.

---

> ### Author Response · Authors · 2025-08-05
> **Follow-up on Rebuttal for Submission 11016**
>
> Dear Reviewer ZBig,
>
> We hope this message finds you well. As the discussion period ends on August 6, 2025, we would greatly appreciate any additional feedback or questions you might have regarding our rebuttal. We are happy to clarify or elaborate on any points as needed before then.
>
> Sincerely,
> NeurIPS 2025 Conference Submission11016 Authors

---

> ### Comment · Reviewer_ZBig · 2025-08-07
>
> Thanks for the detailed response provided by the authors.
>
> Some of my concerns have been addressed. However, I still remain skeptical about the motivation for this task. I can understand that translating thermal images into the RGB domain for downstream vision applications. But I am afraid the inversion process could not provide more information for such applications, nor convenient for RGB-based models. The authors should provide more evidence to prove the advantages of such inversion.
>
> So I decided to keep my rating. Thanks again for the detailed response from the authors.
>
>
> Best Regards,
>
> Reviewer ZBig

---

> ### Author Response · Authors · 2025-08-07
> **Follow-up on Motivation Question for Submission 11016**
>
> We thank Reviewer ZBig for the feedback and would like to address what we believe may be a misunderstanding regarding the core motivation of our work. Specifically, our focus is not on benefiting **RGB-based models**, but rather on enhancing **thermal-based models** and **cross-modality systems**.
>
> While thermal-to-RGB translation is indeed valuable for leveraging existing RGB-model infrastructure, we argue that the reverse task—**RGB-to-thermal translation**—holds distinct and practical merit, especially in scenarios where thermal imaging is unavailable, limited, or cost-prohibitive. Importantly, our goal is not to reconstruct exact thermal readings, but to synthesize reasonable thermal representations that preserves semantic and physical cues otherwise captured by thermal cameras.
>
> We emphasize three primary motivations behind this task:
>
> 1. **Domain Adaptation and Pretraining:** Our approach enables the generation of synthetic thermal data from abundant RGB datasets, facilitating domain adaptation for thermal-based applications such as night-time surveillance, object detection, and navigation. This also enables effective pretraining using synthesized thermal data paired with annotations from RGB-rich datasets, which can subsequently be transferred to real thermal domains.
>
> 2. **Cross-Domain Augmentation:** Synthesized thermal images provide an additional precisely-paired modality for augmenting training datasets in thermal or multi-modal learning settings. This augmentation improves robustness and generalization, particularly in environments where diverse and high-quality thermal data is scarce.
>
> 3. **Cost-Effective Thermal Approximation:** Thermal sensors are often expensive, power-intensive, and subject to usage restrictions. Our method provides a practical alternative by approximating thermal representations from standard RGB imagery, enabling thermal-aware analysis and model prototyping without the need for specialized hardware.
>
> These motivations are further supported by prior work [1–5], which has shown the value of synthetic thermal data in both thermal-specific and cross-modal applications. We hope this clarification more clearly communicates the relevance and potential impact of our approach.
>
> We respectfully request the reviewer to reconsider their evaluation in light of this broader context.
>
> References:
> [1] Mao, F., Mei, J., Lu, S., Liu, F., Chen, L., Zhao, F., & Hu, Y. (2026). PID: physics-informed diffusion model for infrared image generation. Pattern Recognition, 169, 111816.
> [2] Ren, J., Jiang, X., Li, Z., Liang, D., Zhou, X., & Bai, X. (2025). Minima: Modality invariant image matching. In Proceedings of the Computer Vision and Pattern Recognition Conference (pp. 23059-23068).
> [3] He, X., Yu, H., Peng, S., Tan, D., Shen, Z., Bao, H., & Zhou, X. (2025). Matchanything: Universal cross-modality image matching with large-scale pre-training. arXiv preprint arXiv:2501.07556.
> [4] Lee, D. G., Jeon, M. H., Cho, Y., & Kim, A. (2023, May). Edge-guided Multi-domain RGB-to-TIR image Translation for Training Vision Tasks with Challenging Labels. In 2023 IEEE International Conference on Robotics and Automation (ICRA) (pp. 8291-8298). IEEE.
> [5] Xiao, J., Tortei, D., Roura, E., & Loianno, G. (2023, October). Long-range uav thermal geo-localization with satellite imagery. In 2023 IEEE/RSJ International Conference on Intelligent Robots and Systems (IROS) (pp. 5820-5827). IEEE.

---

> > ### Comment · Reviewer_ZBig · 2025-08-08
> >
> > Thanks for the reply from the authors. I still think the contribution and application of this method are limited, so I will maintain my rating.

---

> > > ### Author Response · Authors · 2025-08-08
> > >
> > > We sincerely thank the reviewer for their continued engagement and feedback. However, we respectfully disagree with the assessment that the contributions and applications of our method are limited. RGB-to-thermal image translation is directly applicable to a wide range of impactful real-world tasks, including but not limited to: autonomous driving (e.g., vehicle and pedestrian detection) [1,2], human recognition (e.g., person re-identification [3] and facial landmark analysis [4]), aerial surveillance using UAVs [5], fire detection and monitoring [6,7], and agricultural inspection [8]. These diverse applications demonstrate the broad utility and relevance of our approach. Once again, we appreciate the reviewer’s time and thoughtful evaluation.
> > >
> > > References:
> > > [1] Sikdar, A., Saadiyean, Q., Anand, P., & Sundaram, S. (2024, October). SSL-RGB2IR: Semi-supervised RGB-to-IR Image-to-Image Translation for Enhancing Visual Task Training in Semantic Segmentation and Object Detection. In 2024 IEEE/RSJ International Conference on Intelligent Robots and Systems (IROS) (pp. 5017-5023). IEEE.
> > > [2] Lee, D. G., Jeon, M. H., Cho, Y., & Kim, A. (2023, May). Edge-guided Multi-domain RGB-to-TIR image Translation for Training Vision Tasks with Challenging Labels. In 2023 IEEE International Conference on Robotics and Automation (ICRA) (pp. 8291-8298). IEEE.
> > > [3] Kniaz, V. V., Knyaz, V. A., Hladuvka, J., Kropatsch, W. G., & Mizginov, V. (2018). Thermalgan: Multimodal color-to-thermal image translation for person re-identification in multispectral dataset. In Proceedings of the European conference on computer vision (ECCV) workshops (pp. 0-0).
> > > [4] Flotho, P., Piening, M., Kukleva, A., & Steidl, G. (2025). T-FAKE: Synthesizing Thermal Images for Facial Landmarking. In Proceedings of the Computer Vision and Pattern Recognition Conference (pp. 26356-26366).
> > > [5] Han, Z., Zhang, Z., Zhang, S., Zhang, G., & Mei, S. (2023). Aerial visible-to-infrared image translation: Dataset, evaluation, and baseline. Journal of remote sensing, 3, 0096.
> > > [6] Boroujeni, S. P. H., & Razi, A. (2024). Ic-gan: An improved conditional generative adversarial network for rgb-to-ir image translation with applications to forest fire monitoring. Expert Systems with Applications, 238, 121962.
> > > [7] Li, Y., Ko, Y., & Lee, W. (2023). A feasibility study on translation of RGB images to thermal images: Development of a machine learning algorithm. SN Computer Science, 4(5), 555.
> > > [8] Krestenitis, M., Ioannidis, K., Vrochidis, S., & Kompatsiaris, I. (2025). Visual to near-infrared image translation for precision agriculture operations using GANs and aerial images. Computers and Electronics in Agriculture, 237, 110720.

---

> > > ### Comment · Area_Chair_HFyb · 2025-08-08
> > >
> > > Thanks a lot for your great contributions to NeurIPS. The authors provided further feedbacks regarding the potential contribution/applications. We respect your opinions, so this is just a kind reminder. AC

---

### Official Review · Reviewer_Njbx · 2025-07-03

**Clarity:** 3
**Significance:** 3
**Originality:** 2
**Rating:** 5
**Confidence:** 5

**Summary:**

The paper addressed the scarcity of RGB-thermal imaging data and proposed the ThermalGen model based on a flow generation framework. By integrating an RGB conditional architecture with a style disentanglement mechanism, this model could generate high-fidelity thermal images across different viewpoints, sensors, and environments. Additionally, three new satellite-aerial RGB-T datasets were introduced. In multiple benchmark tests, the model outperformed existing GAN and diffusion models.

**Questions:**

The paper primarily focused on integrating existing datasets with its newly proposed datasets to enable RGB-to-IR translation across different styles, diurnal environments, and viewpoints. However, it did not mention how to disentangle these three factors (style, environment, viewpoint) nor conduct any controlled-variable experiments to verify whether manipulating a single variable could yield realistic translation results. Additionally, the study did not construct a 3D model; instead, it merely translated RGB images from different viewpoints without introducing any novel techniques for viewpoint transformation as the approach for handling viewpoint variations lacks distinctiveness beyond basic image translation. When these issues remain unexplained, the paper’s innovativeness is called into question.

**Ethical Concerns:**

["NO or VERY MINOR ethics concerns only"]

**Final Justification:**

The authors have sufficiently addressed core my concerns. They clarified style embeddings via empirical validations (e.g., dataset-specific comparisons, t-SNE analyses) and demonstrated ThermalGen’s advantages over CLIP-based methods through new experiments. For weak performance on challenging datasets (e.g., low-thermal-contrast scenes), they proposed actionable solutions (CFG scale adjustment) with quantitative improvements. While minor gaps remain (e.g., limited downstream task exploration, no explicit 3D modeling for viewpoints), these are reasonable given the scope, and the authors clearly align their work to dataset - level adaptation rather than strict factor disentanglement. For this conference paper, after carefully reviewing the authors’ rebuttal, I recommend acceptance.

**Limitations:**

yes

**Paper Formatting Concerns:**

I do not notice any major formatting issues in this paper.

**Quality:**

3

**Strengths And Weaknesses:**

Strengths:
1) ThermalGen introduced a flow-based generative model with style-disentangled mechanism, which was the first to explicitly model variations in viewpoints, sensors, and environments in RGB-T translation. The use of adaptive layer normalization (adaLN-Zero) and dataset-specific style embeddings enabled robust generation across diverse styles without retraining.
2) The paper released three new large-scale satellite-aerial RGB-T datasets (DJI-day, BosonPlus-day/night), covering diverse times (day/night), sensor types (Zenmuse H20T, BosonPlus), and geographic regions.

Weaknesses:
1) The style embeddings were vaguely described, as they attributed diurnal environments, sensor styles, and viewpoint variations entirely to datasets without elaborating on how to disentangle these three factors. Moreover, the study did not use the control variable method to verify whether controlling a certain variable could make the generated images change and conform to reality.
2) The paper did not compare with other style transfer methods (such as CLIP-guided embeddings or text-based conditioning methods), which limited the universality of its conclusions. The ablation study on style embeddings (Figure 4 (c)) could further analyze the encoding methods of different styles in the latent space.
3) ThermalGen underperformed on low-thermal-contrast datasets (Boson-night), homogeneous scenes (LLVIP), or under extreme lighting conditions (FLIR). Although the authors attributed these issues to data characteristics, they did not propose specific solutions (such as contrast enhancement modules or domain-specific data augmentation) to mitigate these problems.
minor：
4) The paper failed to specify the computing resources required for training ThermalGen, such as GPU types, training time, and memory usage.
5) The flow-based model relied on the SiT framework, but the paper did not theoretically demonstrate why flow models were more suitable for RGB-T image translation than diffusion models, especially in capturing cross-modal dependencies.
6) The evaluation was mainly limited to image alignment tasks, with fewer insights into downstream applications such as object detection.

---

> ### Author Rebuttal · Authors · 2025-07-29
>
> **Q1: The style embeddings were vaguely described, as they attributed diurnal environments, sensor styles, and viewpoint variations entirely to datasets without elaborating on how to disentangle these three factors. Moreover, the study did not use the control variable method to verify whether controlling a certain variable could make the generated images change and conform to reality.**
>
> We emphasize that the following three dataset comparisons can represent the mentioned variations: Bosonplus-night vs. Boson-night (sensor variation), Bosonplus-day vs. AVIID (viewpoint variation), and Bosonplus-day vs. Bosonplus-night (diurnal variation). Our model consistently demonstrates robust performance across all these datasets (Boson-night issue is mitigated, see Q3), highlighting its capacity to handle such variations effectively.
>
> Our approach uses heterogeneous datasets to learn style embedding specific to each dataset through backpropagation, enabling data-driven learning. These embeddings implicitly capture variations in diurnal conditions, sensor types, and viewpoints without requiring explicit disentanglement. Although we do not enforce factor-wise separation, our empirical results show that ThermalGen generalizes well across diverse domains.
>
> We additionally provide a comprehensive summary of the datasets, categorized for different variations. In the revised manuscript, we include t-SNE analyses of thermal features extracted via DINOv2 across the aforementioned dataset pairs. These comparisons reveal substantial feature discrepancies resulting from varying conditions, yet our model effectively adapts to them.
>
> | Dataset Name | Diurnal Environment | Thermal Sensor Configuration | Viewpoint Variation | Environment Type |
> |--------------|---------------------|-----------------------|---------------------|------------------|
> | Boson-night    | Night           | FLIR Boson | Satellite-aerial     | Wild           |
> | Bosonplus-day    | Day          | FLIR Bosonplus     | Satellite-aerial              | Wild      |
> | Bosonplus-night    | Night            | FLIR Bosonplus     | Satellite-aerial        | Wild      |
> | LLVIP    | Day/Night           | Vanadium Oxide Uncooled Focal Plane Arrays    | Aerial            | Urban          |
> | NII-CU   | Day           | FLIR Vue Pro    | Aerial             | Wild         |
> | AVIID    | Day/Night           | Infrared camera (resolution: 640x480, wavelength:  (8 – 14 μm)    | Aerial             | Urban          |
> | M3FD   | Day/Night           | Infrared camera (resolution: 640x512, wavelength:  (8 – 14 μm))    | Ground           | Diverse          |
> | MSRS   | Day/Night           | InfRec R500    | Ground             | Urban        |
> | FLIR   | Day/Night           | FLIR Tau 2    | Ground             | Urban          |
>
>
> **Q2:The paper did not compare with other style transfer methods (such as CLIP-guided embeddings or text-based conditioning methods), which limited the universality of its conclusions. The ablation study on style embeddings (Figure 4 (c)) could further analyze the encoding methods of different styles in the latent space.**
>
> We conducted a comparative evaluation of ThermalGen using fixed style embeddings against a variant that employs CLIP-based embeddings extracted from RGB images. The fixed style embeddings demonstrate substantial improvements on the following datasets, while maintaining similar performance on other datasets. Additionally, the use of fixed style embeddings enhances computational efficiency by eliminating the need to compute CLIP-based embeddings. These findings highlight both the effectiveness and efficiency of our approach.
> | Methods       |  Bosonplus-day FID ↓| Bosonplus-night FID ↓ | FLIR FID ↓ |
> |----------------|-------------------|---------|-------|
> | ThermalGen-style-embedding | **76.91** | **75.80** | **70.09**  |
> | ThermalGen-CLIP-based-embedding | 106.08 | 89.65 | 75.49 |
>
> **Q3: ThermalGen underperformed on low-thermal-contrast datasets (Boson-night), homogeneous scenes (LLVIP), or under extreme lighting conditions (FLIR). Although the authors attributed these issues to data characteristics, they did not propose specific solutions (such as contrast enhancement modules or domain-specific data augmentation) to mitigate these problems.**
>
> Boson-night: The underperformance can be mitigated by adjusting the classifier-free guidance (CFG) scale factor. The table below illustrates how varying the CFG scale significantly impacts FID performance on this dataset. Notably, a CFG factor of 8.0 yields the best result, reducing the FID to 116.46, which represents a substantial improvement over the default setting.
>
> This finding also underscores the effectiveness of our style embedding mechanism. Since the CFG scale controls the influence of the style embedding during generation, these results demonstrate that appropriately tuning this factor enables the model to generate clearer, higher-fidelity thermal images. Visually, we observe that ThermalGen produces more detailed and realistic outputs at the optimal CFG.
>
> | Method          | Metric       |  CFG Factor = 1.0 (None) | 2.0   | 4.0   | 8.0   | 16.0  |
> |----------------|-------------------|---------|-------|-------|-------|-------|
> | ThermalGen-L/2 | FID ↓             | 161.22  | 157.57| 126.50| **116.46** | 137.95 |
>
> FLIR: We also observe that the underperformance on FLIR can be alleviated by adjusting the CFG scale factor. As shown in the table below, tuning the CFG scale leads to a noticeable reduction in FID, with the best performance achieved at a CFG factor of 4.0.
>
> While the resulting FID does not yet surpass SOTA benchmarks, the trend highlights the potential of our style-disentangled framework to adapt under challenging conditions. The visual results further support this observation, showing improved generation quality as the influence of the style embedding is modulated. These findings suggest a promising direction for enhancing performance in extreme conditions.
>
> | Method          | Metric | CFG Factor = 1.0 (None) | 2.0   | 4.0       | 8.0   |
> |----------------|--------|------------|-------|-----------|-------|
> | ThermalGen-L/2 | FID ↓  | 70.09      | 66.54 | **63.43** | 68.24 |
>
> LLVIP: We conduct t-SNE analysis on DINOv2 features for thermal images and reveal a clear distribution gap between training and testing data, primarily due to camera differences. Expanding the training set is the most effective solution.
>
> **Q4: Computing resources**
>
> Training and evaluation were done on a single NVIDIA H100 or A100 GPU, with both models trained for 200,000 steps. Memory usage (~60 GB) stayed within A100/H100 capacity. Specs will be detailed in the supplementary materials.
>
> **Q5: The paper did not theoretically demonstrate why flow models were more suitable for RGB-T image translation than diffusion models, especially in capturing cross-modal dependencies.**
>
> Our work does not include theoretical proof of the advantage of the flow-based method compared with the diffusion-based method in the RGB-T context. We chose a flow-based method for its faster convergence and strong performance compared with diffusion-based methods, as supported by the Flow Matching paper [1] and SiT’s results [2] (Tables 2–4, Figures 2 and 6). While our study focuses on RGB-T image translation, we assume that these advantages of flow-based methods are generally applicable to image generation tasks, including our own.
>
> **Q6: The evaluation was mainly limited to image alignment tasks, with fewer insights into downstream applications such as object detection.**
>
> Our evaluation does not focus on image alignment tasks but image-to-image translation itself. We agree that ThermalGen has the potential to benefit downstream applications such as image alignment, object detection, and semantic segmentation in the thermal domain. While these are beyond the scope of our current study, they represent promising directions for future work.
>
> **Q7: However, it did not mention how to disentangle these three factors (style, environment, viewpoint) nor conduct any controlled-variable experiments to verify whether manipulating a single variable could yield realistic translation results. Additionally, the study did not construct a 3D model; instead, it merely translated RGB images from different viewpoints without introducing any novel techniques for viewpoint transformation as the approach for handling viewpoint variations lacks distinctiveness beyond basic image translation.**
>
> We would like to clarify that the primary objective of our work is not to explicitly disentangle the three factors—style, environment, and viewpoint—but to learn dataset-level embeddings that capture these attributes in a unified and implicit manner. The style embedding enables our model to adapt to a wide variety of datasets with diverse characteristics, which is demonstrated in our broad evaluation on nine datasets with different attributes (refer to Q1 table).
>
> Regarding viewpoint variation, our method does not involve 3D modeling or introduce novel techniques for explicit viewpoint transformation. In our context, viewpoint variation specifically refers to differences in camera perspectives—such as aerial vs. ground views—rather than geometric transformations within a 3D space. Our results have demonstrated that ThermalGen effectively handles such variation through its style-disentangled, flow-based generative model, which is distinct from basic RGB-T image translation.
>
> References:
> [1] Lipman, Y., Chen, R. T. Q., Ben-Hamu, H., Nickel, M., & Le, M. (2023). Flow Matching for Generative Modeling. The Eleventh International Conference on Learning Representations.
> [2] Ma, N., Goldstein, M., Albergo, M. S., Boffi, N. M., Vanden-Eijnden, E., & Xie, S. (2024, September). Sit: Exploring flow and diffusion-based generative models with scalable interpolant transformers. In European Conference on Computer Vision (pp. 23-40). Cham: Springer Nature Switzerland.

---

> > ### Author Response · Authors · 2025-08-05
> > **Follow-up on Rebuttal for Submission 11016**
> >
> > Dear Reviewer Njbx,
> >
> > We hope this message finds you well. As the discussion period ends on August 6, 2025, we would greatly appreciate any additional feedback or questions you might have regarding our rebuttal. We are happy to clarify or elaborate on any points as needed before then.
> >
> > Sincerely,
> > NeurIPS 2025 Conference Submission11016 Authors

---

### Note · Authors · 2025-08-11

We sincerely thank all reviewers for their time and effort in evaluating our paper. Our work presents a novel **flow-based generative model with a style-disentanglement mechanism** for RGB-to-Thermal image translation, along with dataset curation for large-scale multi-dataset joint training that incorporates **viewpoint, day-night, sensor and environment variation**. In addition, we contribute **three high-quality RGB–thermal datasets** to the research community.

- **Reviewer Njbx** asked for clarification about the style embedding mechanism and suggested a comparison with CLIP-based embeddings. We offered detailed clarification and experiments, but unfortunately, no further discussion followed.
- **Reviewer ZBig** questioned the potential applications and broader contributions of our RGB-to-Thermal translation work; we have responded to these points in detail.
- **Reviewer Njbx** and **Reviewer ZBig** raised concerns regarding performance degradation on certain datasets. We have provided promising solutions or potential directions to mitigate these issues.
- **Reviewer sJDP** requested comparisons with general image-translation baselines; in response, we provided results using the BBDM baseline.
- **Reviewer N5UX** affirmed the soundness of our contributions and expressed a clear inclination toward acceptance.

The responses and suggestions will be incorporated into the revised manuscript. Once again, we appreciate all reviewers’ valuable feedback and constructive discussion, which have helped strengthen our work.

---

### Decision · Program_Chairs · 2025-09-17

**Decision:**

Accept (poster)

**Comment:**

This paper is on translating an RGB image into the thermal domain using flow-based generative models trained on paired RGB–thermal images. The contributions span dataset collection, task-specific algorithm development, and systematic experimental evaluation. After extensive author–reviewer interactions, three reviewers are positive (Accept). Although reviewer ZBig initially raised concerns about the meaningfulness of translating RGB images into thermal images, ZBig raised the score to Borderline Accept. Given the unanimous recommendations by all four reviewers, the AC believes this paper lies above the acceptance bar and recommends acceptance as a poster. Yet it is hard to recommend a spotlight or oral, since the AC believes that the concern raised by reviewer ZBig is quite significant. Considering that thermal imaging is widely used for temperature measurement, the proposed technique will hardly help in this important field, since minor changes in temperature cannot be captured by RGB imaging.